# TMED2 binding restricts SMO to the ER and Golgi compartments

**Giulio Di Minin**[1]*, **Markus Holzner**[1], **Alice Grison**[2], **Charles E. Dumeau**[1], **Wesley Chan**[3,4], **Asun Monfort**[1], **Loydie A. Jerome-Majewska**[3,4], **Henk Roelink**[5], **Anton Wutz**[1]*

**1** Institute of Molecular Health Sciences, Department of Biology, Swiss Federal Institute of Technology ETH Hönggerberg, Zurich, Switzerland, **2** Department of Biomedicine, University of Basel, Basel, Switzerland, **3** Department Anatomy and Cell Biology, Human Genetics and McGill University, Montreal, Canada, **4** Department of Pediatrics, Human Genetics and McGill University, Montreal, Canada, **5** Department of Molecular and Cell Biology, University of California, Berkeley, California, United States of America

* giulio.diminin@biol.ethz.ch (GDM); awutz@ethz.ch (AW)

**Data Availability Statement:** All relevant data are within the paper and its Supporting Information files. The sequencing datasets are deposited in the NCBI Short-Read Archive (http://www.ncbi.nlm.

## Abstract

Hedgehog (HH) signaling is important for embryonic pattering and stem cell differentiation. The G protein–coupled receptor (GPCR) Smoothened (SMO) is the key HH signal transducer modulating both transcription-dependent and transcription-independent responses. We show that SMO protects naive mouse embryonic stem cells (ESCs) from dissociation-induced cell death. We exploited this SMO dependency to perform a genetic screen in haploid ESCs where we identify the Golgi proteins TMED2 and TMED10 as factors for SMO regulation. Super-resolution microscopy shows that SMO is normally retained in the endoplasmic reticulum (ER) and Golgi compartments, and we demonstrate that TMED2 binds to SMO, preventing localization to the plasma membrane. Mutation of TMED2 allows SMO accumulation at the plasma membrane, recapitulating early events after HH stimulation. We demonstrate the physiologic relevance of this interaction in neural differentiation, where TMED2 functions to repress HH signal strength. Identification of TMED2 as a binder and upstream regulator of SMO opens the way for unraveling the events in the ER–Golgi leading to HH signaling activation.

## Introduction

Hedgehog (HH) signaling controls key events in embryonic development, tissue homeostasis, and repair [1–3]. Deregulation of HH signaling by genetic or pharmacologic means causes severe developmental abnormalities, and activation of the HH response is frequently implicated in tumor initiation and dissemination [4]. Compounds inhibiting the HH signal transducer Smoothened (SMO) are used for therapy of basal cell carcinoma (BCC) and medulloblastoma [5].

The HH receptor Patched (PTCH) inhibits SMO, and binding of Sonic, Indian, or Desert HH (SHH, IHH, and DHH, respectively) releases this inhibition [6]. SMO activation can have both transcriptional and nontranscriptional effects. SMO is a G protein–coupled receptor (GPCR) and regulates cAMP levels and cytoskeleton dynamics [7–9], as well as calcium levels

nih.gov/sra) and can be accessed using accession numbers SRR8449980 and SRR8449981.

**Funding:** GDM was supported by the ETH Zurich Postdoctoral Fellowship Program as well as the Marie Curie Actions for People COFUND Program. WC and LAJM were supported by a grant from the Natural Sciences and Engineering Research Council of Canada (RGPIN-2015-06699). LJM is a member of the Research Centre of the McGill University Health Centre which is supported in part by FRQS. HR was supported by a grant from NIGMS (R01GM117090). This work was supported by grants from the Swiss National Science Foundation (SNF grants 31003A_152814/1 and 31003A_175643/1) to AW. The funders had no role in study design, data collection and analysis, decision to publish, or preparation of the manuscript.

**Competing interests:** The authors have declared that no competing interests exist.

**Abbreviations:** BCC, basal cell carcinoma; bFGF, basic fibroblast growth factor; CaSR, calcium sensing receptor; CBC, coordinate-based colocalization; co-IP, co-immunoprecipitation; COPI, coat protein complex I; DHH, Desert hedgehog; EE, early endosome; EGF, epidermal growth factor; EGFR, epidermal growth factor receptor; ER, endoplasmic reticulum; ESC, embryonic stem cell; FBS, fetal bovine serum; GPCR, G protein–coupled receptor; gRNA, guide RNA; HA, hemagglutinin; HH, hedgehog; IHH, Indian hedgehog; LAMPCR, linear amplification PCR; LE, late endosome; LIF, leukemia inhibitory factor; MEF, mouse embryonic fibroblast; MTOC, microtubule organizing center; NGS, next generation sequencing; NPC, neural progenitor cell; NPC1, Niemann–Pick C1; PMP, purmorphamine; PTCH, Patched; PTX, pertussis toxin; RA, retinoic acid; RT-qPCR, quantitative reverse transcription PCR; SAG, smoothened agonist; SHH, Sonic hedgehog; siRNA, small interfering RNA; SMO, Smoothened.

affecting cell metabolism in muscle and brown fat [10]. The transcriptional response to SMO activation is mediated by the GLI family of zinc finger transcription factors. In vertebrates, activation of GLI proteins appears to be limited to a specialized plasma membrane compartment, the primary cilium [11,12]. In mice, a $Gli1/Gli2^{-/-}$ double mutation or primary cilium deficiencies [13,14] result in milder phenotypes than a $Shh/Ihh$ double mutation [15] and a $Smo$ mutation [15]. These phenotypic differences suggest the relevance of GLI independent functions of SMO.

PTCH1 is a member of the RND family of proton-driven antiporters [16], which is conserved in all domains of life. PTCH1 shares characteristics with the RND cholesterol transporter Niemann–Pick C1 (NPC1) including a sterol-sensing domain and multiple cholesterol binding sites [17–22]. In its unliganded state, PTCH1 affects SMO distribution and prevents its translocation to the primary cilium [23–26], where activation of GLI transcription factors occurs. However, G protein regulation by SMO is not restricted to the primary cilium [27–30]. The primary cilium is dispensable or even inhibitory for nontranscriptional effects of SMO [8]. These findings suggest that different effects of SHH signals (GLI mediated or non-GLI mediated) are determined by SMO localization prior to, or cycling through, the primary cilium.

Screens in cell lines for genes involved in the HH response have been based on a transcriptional (GLI mediated) readout of pathway activation [26,31–33]. Earlier studies have uncovered details of the assembly and trafficking of the primary cilia. Despite impressive progress in understanding events downstream of SMO that has been made, the molecular mechanism of SMO regulation upstream of the primary cilium remains less well understood.

SMO is cotranslationally imported into the endoplasmic reticulum (ER) membrane [34,35]. Transport from the ER through the Golgi to the plasma membrane is highly regulated [36]. The secretory pathway includes bidirectional transport between the ER and the Golgi through coat protein complex I (COPI)-coated and COPII-coated vesicles [37]. Retrograde transport is important for quality control of folding and glycosylation of membrane proteins before they reach the plasma membrane. The precise route of SMO to the plasma membrane, as well as interacting partners determining its localization, remains to be understood [38]. It is thought that SMO reaches the PM passing normally through the Golgi. However, the oncogenic SMO-A1 variant might reach the primary cilium through a different and potentially direct route [38].

Here, we discover that SMO sustains embryonic stem cell (ESC) survival in a GLI-independent manner. We use this dependency for a genetic screen in haploid ESCs for factors involved in the earliest events of SMO activation. Identification of the COPI and COPII components TMED2 and TMED10 allowed to uncover a new mechanism of SMO trafficking from the ER–Golgi apparatus in the regulation of HH signaling.

## Results

### SMO counteracts dissociation-induced apoptosis in ESCs

We observed that $Smo^{-/-}$ ESCs generated by CRISPR/Cas-9 (S1A Fig) are characterized by a lower growth rate compared to control cells. This phenotype was particularly pronounced in a chemically defined culture medium that maintains ground state pluripotency of ESCs [39] (S1B Fig). After passaging, control ESCs attached and spread on the cell culture plate (Fig 1A). In contrast, $Smo^{-/-}$ ESCs became motile, displayed blebbing of the plasma membrane (Fig 1B), and ultimately formed apoptotic bodies. In human ESCs, RHO-ROCK–dependent myosin hyperphosphorylation is the primary cause of apoptosis after dissociation [40,41], which can be circumvented by inhibitors of ROCK kinase. We find that ROCK inhibition also protected

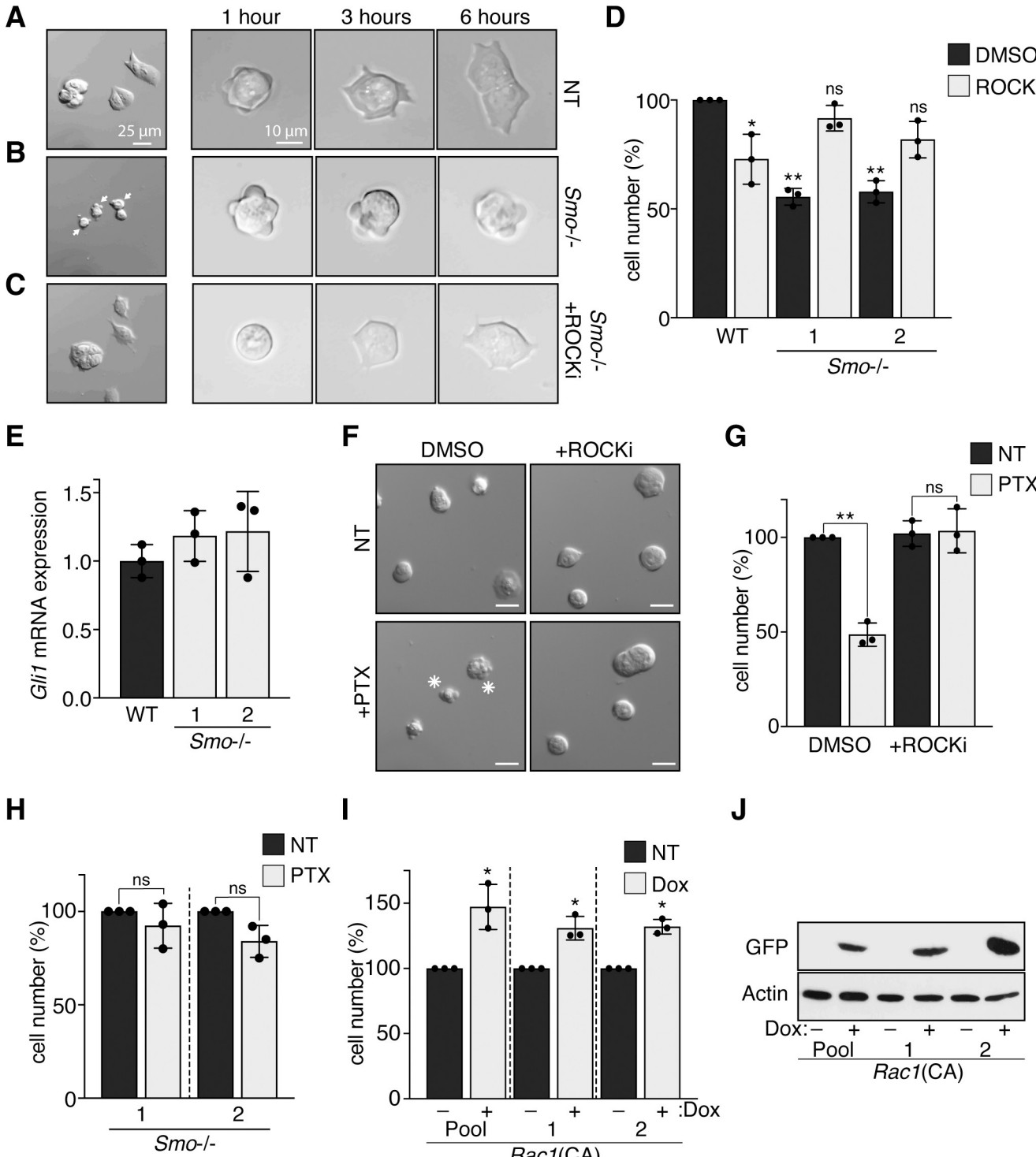

**Fig 1. SMO supports survival of ESCs after dissociation. (A–C)** SMO mutation induces blebbing and death after dissociation of ESCs. On the left, ESCs morphology 24 hours after dissociation and plating on Matrigel (A, untreated; B, pretreated with PMP for 24 hours; C, pretreated with PMP for 24 hours and with ROCKi after plating). Cells showing membrane blebbing (arrow), and apoptotic bodies (asterisk) are indicated. Images at 1, 3, and 6 hours after plating are shown on the right. **(D)** Survival of $Smo^{-/-}$ ESCs after 48 hours with or without addition of ROCKi. Asterisks denote statistical significance for difference from the untreated sample. **(E)** GLI transcription is not affected by $Smo$ deletion in ESCs. $Gli1$ mRNA levels were measured by qPCR. Samples do not show statistically significant differences ($n$ = 3, biological replicates). **(F–H)** $G_i$ protein inhibition induces apoptosis in ESCs. (F) ESC morphology 24 hours after dissociation. Cells were pretreated with the $G_i$ specific inhibitor PTX and ROCKi. Apoptotic bodies are marked by an asterisk. Bars represent

20 μm. (G) Survival of ESCs after 48 hours treatment with PTX with or without the addition of ROCKi. (H) Survival of $Smo^{-/-}$ ESCs after 48-hour treatment with PTX. Two independent clones are shown. **(I, J)** $Rac1$(CA) increases survival after passaging SMO$^{-/-}$ ESCs. A Rac1(CA)-IRES-GFP construct under the control of a Dox-inducible promoter was integrated in ESCs. (I) The survival of the transfected cell pool and of 2 independent clones was analyzed with and without Dox induction for 48 hours. Cell survival was normalized to uninduced cells (NT). Asterisks denote statistical significance for difference from the uninduced sample. (J) Immunoblot analysis showing the induction of the Rac1(CA)-IRES-GFP construct after Dox treatment. The data underlying all the graphs shown in the figure are included in the S1 Data file. Dox, doxycycline; ESC, embryonic stem cell; NT, not treated; PMP, purmorphamine; PTX, pertussis toxin; qPCR, quantitative PCR; SMO, Smoothened.

$Smo^{-/-}$ ESCs from dissociation-induced apoptosis (Fig 1C and 1D). This observation suggested an unanticipated role of SMO in preventing death of pluripotent cells when cell–cell contact is interrupted.

To clarify the role of HH signaling in ESCs, we analyzed GLI transcriptional activity by quantitative reverse transcription PCR (RT-qPCR). Genetic and chemical inhibition of SMO did not perturb the expression of the HH target $Gli1$ and $Ptch1$ (Fig 1E, S1C and S1D Fig). SHH or chemical agonists neither promoted $Gli1$ mRNA expression (S1D Fig). The observation that GLI activity is not responsive to HH signaling correlates with the absence of primary cilia in the majority of ESCs (S1G and S1H Fig), as also previously shown [42]. Additionally, we verified that inhibition of GLI transcription activity using the specific inhibitor GANT61 (S1F Fig) was not sufficient to induce cell death in ESCs (S1E Fig).

SMO can regulate heterotrimeric G proteins of the $G_i$ subclass [43–46]. Direct inhibition of $G_i$ proteins by pertussis toxin (PTX) affected normal spreading of ESCs after dissociation and decreased survival (Fig 1F and 1G). The effects of PTX were rescued by ROCK inhibition. Importantly, PTX did not have a measurable effect on $Smo$ mutant ESCs (Fig 1H). Taken together, these data indicated that $G_i$ proteins might be involved in SMO function in naive mouse ESCs. $G_i$ protein modulation by SMO has previously been implicated in RHOA and RAC1 regulation of the ROCK1 kinase [47]. We observe that inducible expression of a constitutive active RAC(Q61L) increased the survival of $Smo$ mutant ESCs (Fig 1I and 1J), further supporting the view that SMO counteracts ROCK1-induced actin–myosin contractility in mouse ESCs.

## High purmorphamine doses induce ESC death by counteracting SMO

Our observations of a new function of SMO for sustaining mouse ESCs survival led us to further investigate chemical agonists and inhibitors of SMO. SMO antagonists have been classified based on their repression of GLI transcription. Their effect on GLI-independent SMO functions is less clear. Cyclopamine has been reported to either counteract [7] or sustain [10] SMO GPCR activity dependent on the cellular context. Smoothened agonist (SAG) and Purmorphamine (PMP) (Fig 2A) have a biphasic bell-shaped activity curve where they activate GLI-dependent transcription at low but become inhibitory at high concentrations [48–50]. We observed that the HH antagonists KAAD-cyclopamine and SANT1 did not affect ESC survival (Fig 2B) at concentrations that inhibit GLI transcriptional activity in NIH-3T3 cells (S2A Fig). Recombinant SHH did also not affect ESC viability (Fig 2B) at concentrations that induced a strong GLI transcriptional response in NIH-3T3 cells (S2A Fig). Similarly, low concentrations of the agonists SAG and PMP had no effect. In contrast, high concentrations of SAG (2.5 to 5 μM) and PMP (5 to 10 μM) strongly impaired the survival of ESCs (Fig 2B). Similar concentrations did not decrease NIH-3T3 cell survival (S2B Fig). Cytotoxicity of SMO agonists in ESCs was surprising and required PMP concentrations above 2.5 μM (S2C Fig), which is an order of magnitude higher than the concentration used to activate SMO [48–50]. We verified that high PMP concentrations repress GLI transcription in NIH-3T3 cells and $Ptch1^{-/-}$ mouse embryonic fibroblasts (MEFs). We measure an IC50 approximately 4 μM for PMP in

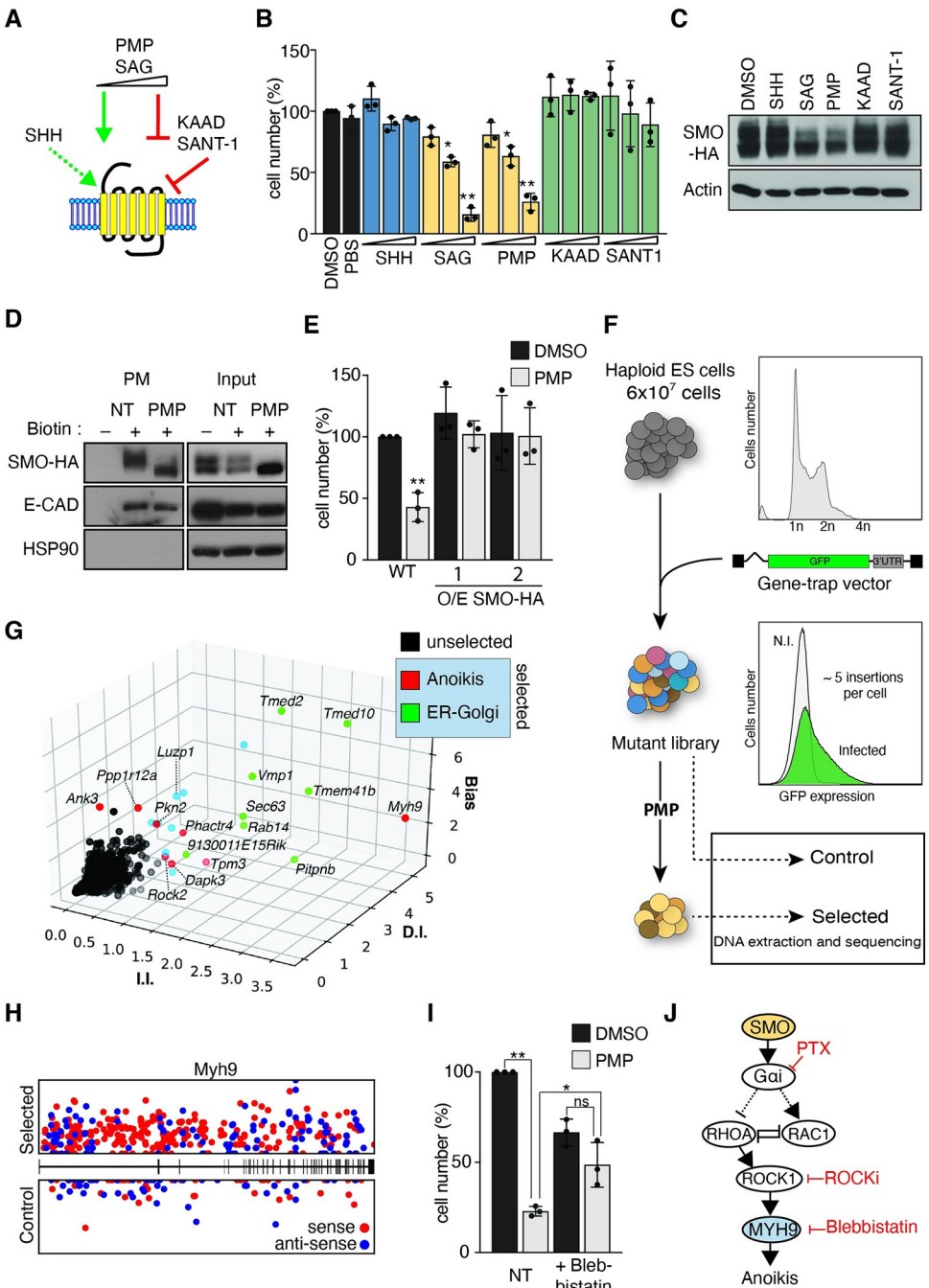

**Fig 2. Genetic screen in haploid ESCs for PMP resistance identifies Anoikis modulators. (A)** Model of the activity of known SMO targeting compounds. **(B)** SAG and PMP decrease survival of ESCs treated for 48 hours with SHH (50 to 500 ng/ml), SAG (0.5 to 5 μM), PMP (1 to 10 μM), Cyclopamine-KAAD (1 to 5 μM), and SANT-1 (10 to 50 μM). Cell counts are normalized to DMSO treated sample. **(C, D)** High PMP concentrations affect SMO protein levels. (C) ESCs expressing SMO–HA were treated for 24 hours with the indicated compounds, and SMO levels were analyzed by immunoblot using an anti-HA antibody. Actin is shown as loading control. (D) Cells were treated with NHS-SS-Biotin (Biotin) to label PM proteins and with high PMP concentrations as indicated. Western analysis of PM proteins and input before purification (1/50 of pull-down) are shown. SMO levels were analyzed by immunoblot using an anti-HA antibody. E-CAD and HSP90 are shown as loading control and to confirm PM protein purification. **(E)** SMO–HA overexpression confers resistance to PMP. Survival of ESCs after PMP treatment (5 μM). The effect on 2 independent clones described in S2J Fig is shown. Asterisks denote statistical significance for difference from the WT DMSO treated sample. **(F)** Schematic screening strategy for PMP resistance. **(G)** Identification of genes conferring PMP resistance by enrichment of I.I., D.I., and gene trap orientation bias (Bias). Top candidates are marked in light blue. Selected

candidates implicated in anchorage independent growth (red) or in ER–Golgi trafficking (green) are marked and annotated. **(H)** Distribution of I.I. within *Myh9* in PMP selected (top) and control (bottom) samples. Gene trap insertions in the orientation of the gene transcription unit (sense) are marked in red, and antisense insertions are marked in blue. **(I)** The MYH9 inhibitor blebbistatin mediates resistance to PMP. Survival of ESCs treated for 48 hours with PMP with or without the addition of blebbistatin. Asterisks denote statistical significance for difference between indicated samples. **(J)** Model summarizing the mechanism of SMO function in sustaining ESC survival. Compounds are annotated in red. The data underlying all the graphs shown in the figure are included in the S1 Data file. D.I., disrupting insertion; ER, endoplasmic reticulum; ESC, embryonic stem cell; HA, hemagglutinin; I.I., independent insertion; NT, not treated; PM, plasma membrane; PMP, purmorphamine; PTX, pertussis toxin; SAG, smoothened agonist; SHH, Sonic hedgehog; SMO, Smoothened; WT, wild-type.

NIH-3T3 cells (S2D Fig). In $Ptch1^{-/-}$ MEFs, the absence of PTCH1 leads to constitutive activation of GLI (S2E Fig). Addition of PMP at concentrations higher than 2.5 μM strongly repressed GLI transcription in $Ptch1^{-/-}$ MEFs. Conversely, in $Smo^{-/-}$ ESCs, SAG and PMP treatment did not lead to elevated cell death compared to untreated controls (S2F Fig). PMP treatment further induced membrane blebbing and phosphorylation of MYL2 at the cell cortex in wild-type ESCs consistent with a role of myosin hypercontractility in PMP induced cell death (S2G and S2I Fig). ROCK inhibition or expression of constitutive active RAC1 conferred resistance to PMP (S2H and S2J and S2K Fig), indicating that high concentrations of SAG and PMP decrease ESC survival similar to a *Smo* mutation.

## Purmorphamine reduces SMO protein abundance in ESCs

To investigate how GLI-independent SMO signaling activity is inhibited by high concentrations of PMP and SAG, we decided to analyze SMO protein level and distribution in ESCs. We stably introduced a cDNA construct for expression of a carboxyl-terminally hemagglutinin (HA) epitope tagged SMO protein (SMO–HA) into $Smo^{-/-}$ cells. We observed that treatment with high concentrations of PMP or SAG led to a decrease in SMO, particularly the higher molecular weight form, using western analysis (Fig 2C). SMO glycosylation in the ER and Golgi causes an increased molecular weight [51]. In PMP-treated cells, these higher molecular weight forms of SMO became undetectable at the plasma membrane (Fig 2D). Our observations suggest that loss of glycosylated SMO from the plasma membrane is correlated with cell death. Notably, it has been shown that SMO glycosylation even if dispensable for GLI regulation is required for modulating G protein activity [52]. We investigated if high expression of SMO would rescue PMP induced cell death. We introduced the SMO–HA construct into wild-type ESCs and obtained high expressing clones (S2L Fig). Importantly, SMO overexpression conferred PMP resistance to mouse ESCs (Fig 2E).

## Screening for resistance to SMO selective compounds in mouse haploid ESCs

Our data uncovered a function of SMO in ESC survival and provided an unexpected opportunity for forward genetic screening of new factors that regulate SMO. Haploid ESCs have been previously shown to be suitable for efficient screening of developmental pathways [53–55]. Treatment of ESCs with high PMP concentrations induces cell death through a loss of SMO function allowing to select for survival mutations.

We infected 60 million haploid ESCs with a viral gene trap vector to obtain a genome-wide library of mutations (Fig 2F). Three independent mutant pools were subsequently split into 2 populations that were either selected with PMP or used as a control. We observed a PMP-resistant population after 12 days (S3A Fig). Over 2 million independent viral insertions were identified by next generation sequencing (NGS) in control and selected samples. Insertions were

distributed over all chromosomal regions and showed an expected bias in transcribed regions (S3B and S3C Fig). Candidate gene prediction was performed using the HaSAPPy package [56] (Fig 2G, S3D and S3E Fig). Genes associated with GLI activation or primary cilium were not discovered likely reflecting cell system and selection strategy of our screen (S1 Table). Selected genes separated into 2 categories associated with cytoskeleton and ER–Golgi (Fig 2G, S3D and S3E Fig, S1 Table). The first category contained genes functioning in anchorage independent growth, whose mutations have been implicated in resistance to Anoikis-induced cell death in tumors (S3F Fig). We detected strong evidence for selection of inactivating mutations of *Myh9* in PMP selected ESCs (S1 Table, Fig 2H). Hyperactivation of MYH9 has also been implicated in dissociation-induced apoptosis in human ESCs [40,41]. The MYH9 inhibitor blebbistatin has been shown to increase survival when single cell suspensions of human ESCs are prepared. We find that treatment with blebbistatin similarly rescued mouse ESCs from PMP induced death (Fig 2I). This observation is therefore consistent with our earlier finding that SMO increases RAC1 activity and counteracts ROCK1-induced myosin contractility in naive mouse ESCs (Fig 2J).

## The Golgi protein TMED2 modulates HH signaling

The second group of candidates comprises proteins that localize in the Golgi and function in vesicle trafficking (Fig 2G, S3E Fig, S1 Table). Among them, we selected *Tmed2* and *Tmed10*, which showed the strongest evidence for selection (Fig 3A). Both are members of the p24 family of cargo receptors [57]. Mutations in other members of the p24 family were not enriched in our screen (S4A Fig). To further characterize their function, we established *Tmed2* and *Tmed10* mutant ESCs using CRISPR/Cas-9 nucleases (S4B Fig). Western analysis confirmed the absence of protein in *Tmed2* and *Tmed10* mutant ESC lines (Fig 3B). In addition, *Tmed10* mutant ESCs showed a strong reduction of TMED2 protein. This observation demonstrated a dependence of TMED2 on TMED10 in mouse ESCs, which is consistent with interdependence of p24 family proteins as observed before [58–60]. Loss of *Tmed2* and *Tmed10* did not impair self-renewal of ESCs (S4C and S4D Fig). Furthermore, ER, Golgi, and late endosome (LE) vesicles appeared similar in *Tmed2* mutant and wild-type ESCs (Fig 3C–3E). *Tmed2* and *Tmed10* mutant ESCs were highly resistant to PMP when compared to control ESCs (Fig 3F). In addition, PMP induced cortical MYL2 phosphorylation was not observed in *Tmed2* mutant ESCs (S2I Fig). We performed western analysis of SMO–HA after PMP treatment. PMP caused a loss of the high molecular weight form of SMO in control ESCs (Fig 3G). In contrast, the decrease was moderated in *Tmed2* mutant ESCs. Resistance of *Tmed2*$^{-/-}$ ESCs to PMP thus correlates with an increase in glycosylated SMO.

## TMED2 is a negative regulator of HH signaling in neural differentiation

To further investigate a function of TMED2 in HH signaling, we measured GLI transcriptional activity in NIH-3T3 cells. We depleted *Tmed2* by small interfering RNA (siRNA) transfection and observed a significantly increased *Gli1* (Fig 4A) and *Ptch1* expression (S5A Fig) in response to SHH ligand. The structure of the primary cilia (S5B Fig) and recruitment of SMO to the cilium (Fig 4B, S5C Fig) in *Tmed2*-depleted and control cells were comparable. *Tmed2* depletion did not increase basal GLI transcription in the absence of SHH ligand.

To address the physiological relevance, we investigated the role of TMED2 in neural tube patterning, where HH signaling has an important role for specifying neuronal subtypes [61]. Immunofluorescence staining showed TMED2 expression in neural tube sections of mouse E9.5 embryos that overlapped with Golgi markers (Fig 4C, S5D and S5E Fig). The *Tmed2* mutation in mice leads to embryonic lethality before midgestation, whereby abnormalities as

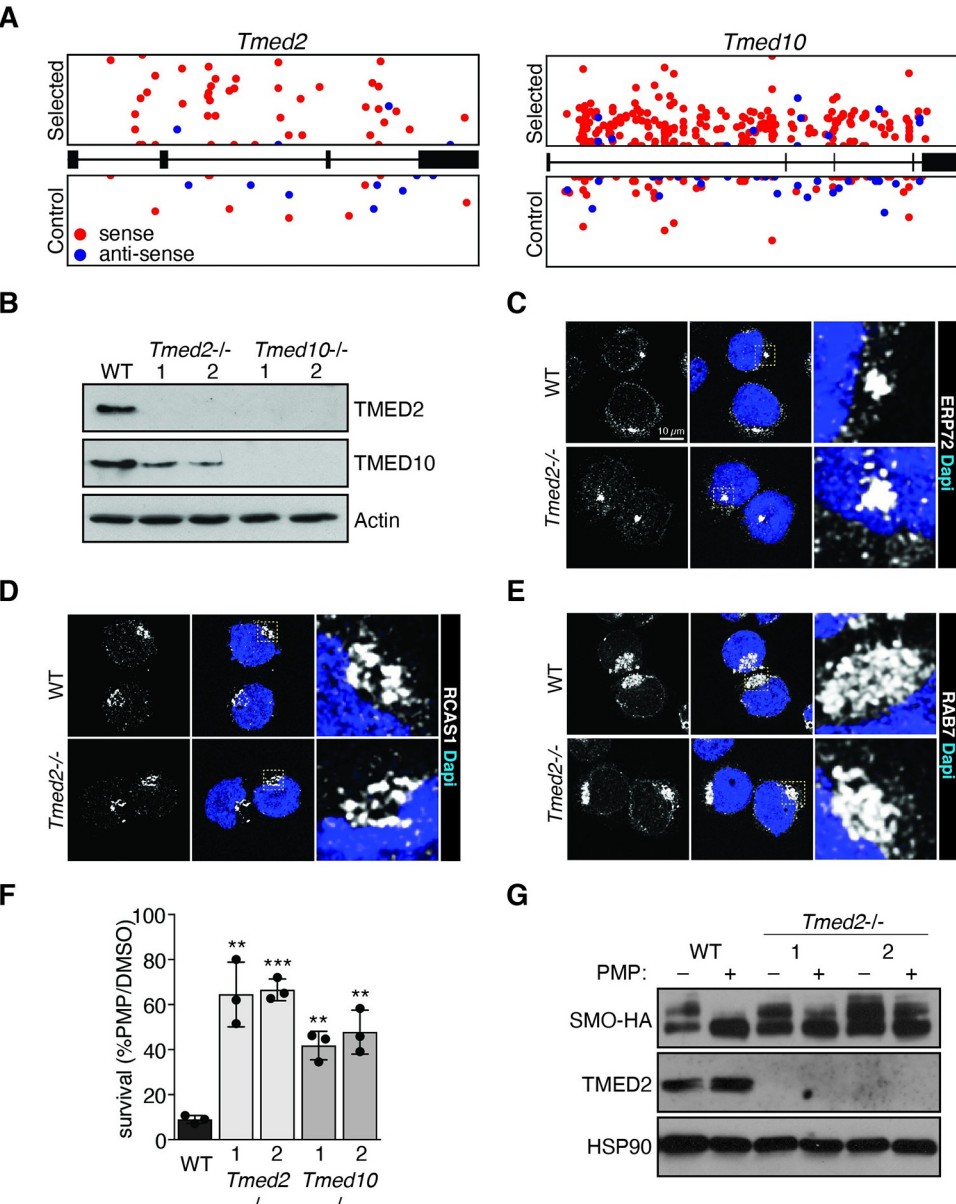

**Fig 3. Identification of TMED2 as a new modulator of HH signaling. (A)** Distribution of I.I. within *Tmed2* and *Tmed10* in PMP selected (top) and control (bottom) samples. **(B)** Western analysis of parental control (WT) and 2 clones of *Tmed2* and *Tmed10* mutant ESCs. **(C–E)** Immunofluorescence staining (C) ERP72 (ER marker), (D) RCAS1 (Golgi marker), and (E) RAB7 (LE marker) in WT and *Tmed2*⁻/⁻ ESCs. Scale bar = 10 μm. Samples were analyzed using conventional FM. **(F)** *Tmed2* and *Tmed10* mutations confer PMP resistance. Survival of *Tmed2* and *Tmed10* mutant ESCs after PMP treatment. Asterisks denote statistical significance for difference from the WT sample. **(G)** SMO–HA protein levels in WT and *Tmed2*⁻/⁻ ESCs. Western analysis of cells treated with or without PMP for 24 hours. The data underlying all the graphs shown in the figure are included in the S1 Data file. ER, endoplasmic reticulum; ESC, embryonic stem cell; FM, fluorescence microscopy; HA, hemagglutinin; HH, hedgehog; I.I., independent insertion; LE, late endosome; PMP, purmorphamine; SMO, Smoothened; WT, wild-type.

well as a developmental delay arise from E7.5 [62]. The developmental delay makes comparison between *Tmed2* mutant and wild-type embryos difficult. However, neural progenitor cells (NPCs) could form normally from *Tmed2* mutant ESCs, allowing to study TMED2 function in patterning in culture. We analyzed the expression of the ventral neural tube marker genes

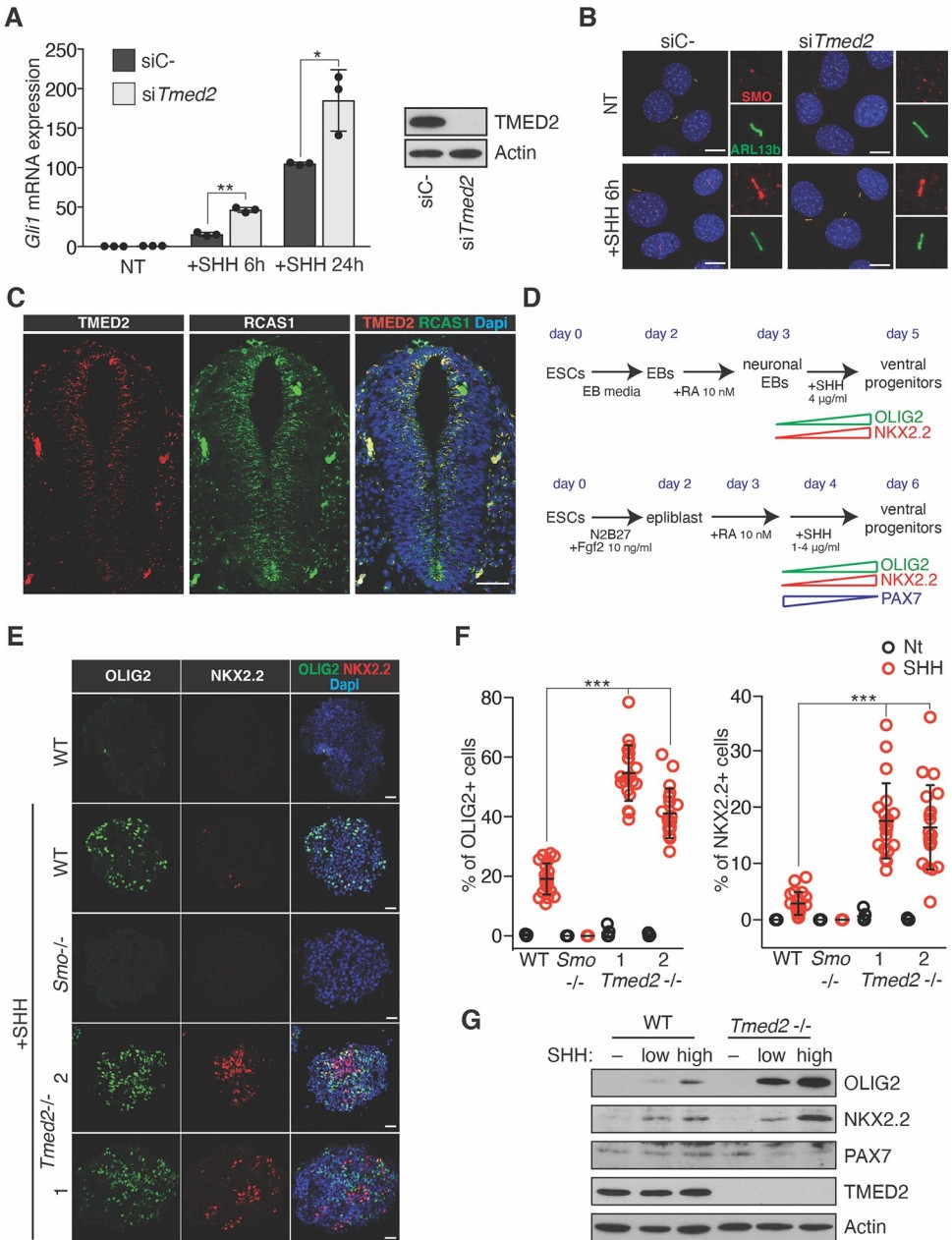

**Fig 4. TMED2 is a negative regulator of HH signaling in neural differentiation. (A, B)** *Tmed2* knockdown promotes GLI activity after SHH treatment in NIH-3T3 cells. NIH-3T3 cells were transfected with an siRNA targeting *Tmed2* (si*Tmed2*) or a nontargeting control (siC-) and treated with SHH for 6 and 24 hours. (A) RT-qPCR of *Gli1* mRNA. Asterisks denote statistical significance for difference between indicated samples. On the left, immunoblot showing siRNA knockdown efficiency of TMED2. Actin is shown as a loading control. (B) Immunofluorescence staining of SMO (red) and ARL13B (green) to visualize primary cilia after 6 hours of SHH treatment. Scale bar = 10 μm. Inserts (right) magnify SMO and ARL13B localization at the primary cilium. **(C)** Immunostaining of TMED2 (red) and the Golgi marker RCAS1 (green) in neural tube sections of WT E9.5 mouse embryos. Scale bar = 50 μm. **(D–G)** *Tmed2* mutation increases SHH-dependent ventral marker expression in NPCs. (D) Schematic overview of the protocol for deriving neuralized EBs from ESCs (top) and of the NPC adherent culture differentiation (AD) protocol (bottom). The expected timing of NPC marker expression is indicated. (E) Representative immunofluorescence images of ventral markers OLIG2, and NKX2.2 in neuralized EBs of indicated genotypes treated with or without SHH. Scale bar = 25 μm. Asterisks denote statistical significance for difference between indicated samples. (F) Percentage of cells expressing OLIG2 (left) and NKX2.2 (right) relative to total cell count in neuralized EBs as in C. Individual EBs are plotted (*n* = 20). (G) Western analysis of ventral markers OLIG2 and NKX2.2 and the dorsal marker PAX7 in WT and *Tmed2*$^{-/-}$ NPCs on day 5. The data underlying all the graphs shown in the figure are

included in the S1 Data file. EB, embryoid body; HH, hedgehog; NPC, neural progenitor cell; RT-qPCR, quantitative reverse transcription PCR; SHH, Sonic hedgehog; siRNA, small interfering RNA; SMO, Smoothened; WT, wild-type.

OLIG2 and NKX2.2 in NPCs derived by aggregation of ESCs into neuralized embryoid bodies (Fig 4D). In the presence of SHH, the number of NPCs expressing ventral markers was increased in *Tmed2*$^{-/-}$ compared to wild-type NPCs (Fig 4E and 4F). A similar ventralizing effect was observed using an adherent culture differentiation (AD) system (Fig 4D and 4G). Conversely, the dorsal marker PAX7 was decreased in *Tmed2* mutant NPCs. Our results indicated that the *Tmed2* mutation enhanced the effects of HH signaling in neural differentiation.

These findings encouraged us to further corroborate if the situation in vivo is at least consistent with our results in culture. During neural tube development, HH is expressed from the notochord and floor plate and induces NKX6.1 in ventral NPCs. Subsequently, HH signaling drives the specification of p3 and pMN progenitor cells coexpressing NKX2.2 and OLIG2, respectively. Higher SHH signaling in proximity to the floor plate induces NKX2.2. In order to assess the role of TMED2, we compared the expression of NKX2.2 and OLIG2 in wild-type and *Tmed2* mutant embryos. At E10.5, the neural tubes of control embryos appeared twice to 3 times larger than that of *Tmed2*$^{-/-}$ embryos (S6A Fig). To account for the developmental delay, we additionally compared sections of E10.5 *Tmed2*$^{-/-}$ embryos to E9.0 and E9.5 control embryos (S6A Fig). E10.5 *Tmed2*$^{-/-}$ neural tubes were similar in size to E9.0 controls, although their dorsal wall still appeared a bit thinner. *Tmed2*$^{-/-}$ embryos showed a OLIG2 domain that appeared shifted dorsally relative to E9.0 controls (S6A Fig). This observation could be caused by a higher sensitivity of *Tmed2* mutant cells to HH signals or the developmental time difference between E9 and E10.5. We therefore investigated the ratio between NKX6.1 and OLIG2 positive cells, which appeared independent of the developmental stage in wild-type embryos between E9 and E10.5. We observed an extensive domain of NKX6.1 expression that contained a confined region of OLIG2 expression in E9 control embryos (Fig 5A–5C, S6C Fig, S2 Table). A similar ratio between OLIG2+ and NKX6.1+ progenitors was also observed in E10.5 control embryos (Fig 5C, S6B Fig). In *Tmed2*$^{-/-}$ embryos, an increased fraction of progenitors coexpressed OLIG2 and NKX6.1 (Fig 5A–5C, S6C Fig). Furthermore, the dorsal marker PAX7 appeared more restricted to a dorsal region of the neural tube in *Tmed2*$^{-/-}$ embryos and covered an extensive region in controls (Fig 5E, S6F Fig). Consistent with this, we also observed an expansion of the intermediate DBX1 expression domain in *Tmed2* mutants (Fig 5D, S6D Fig). This led to a direct juxtaposition of the DBX1 and NKX6.1 domains, whereas in wild-type neural tubes, these domains were separated by a wide gap (S6E Fig). Although these observations are consistent with our results in NPC differentiation that HH effects are strengthened in the absence of TMED2, we cannot rule out other signaling pathways also contribute to the *Tmed2* phenotype. We note that the increase in HH signaling in the absence of TMED2 is different from ectopic activation caused by the absence of the negative modulators PTCH1 and SUFU [63], but appears comparable to a loss of *Gli3* repressor activity [64].

## TMED2 biochemically binds SMO in the ER–Golgi compartment

The amount of SMO at the plasma membrane is a major determinant of the strength of G protein–and GLI-mediated effects. TMED2 has been implicated in vesicle trafficking and protein secretion [59,65–68], which suggested a possible role in controlling the amount of SMO at the plasma membrane. To test this hypothesis, we first analyzed SMO distribution in NPCs. We differentiated *Smo*$^{-/-}$ ESCs transgenically expressing the HA tagged SMO (SMO–HA) for 2 days in N2B27 media and costained with markers of different cellular compartments (S7A and S7B Fig). As previously reported in multiple cellular systems [34], SMO was in juxtanuclear

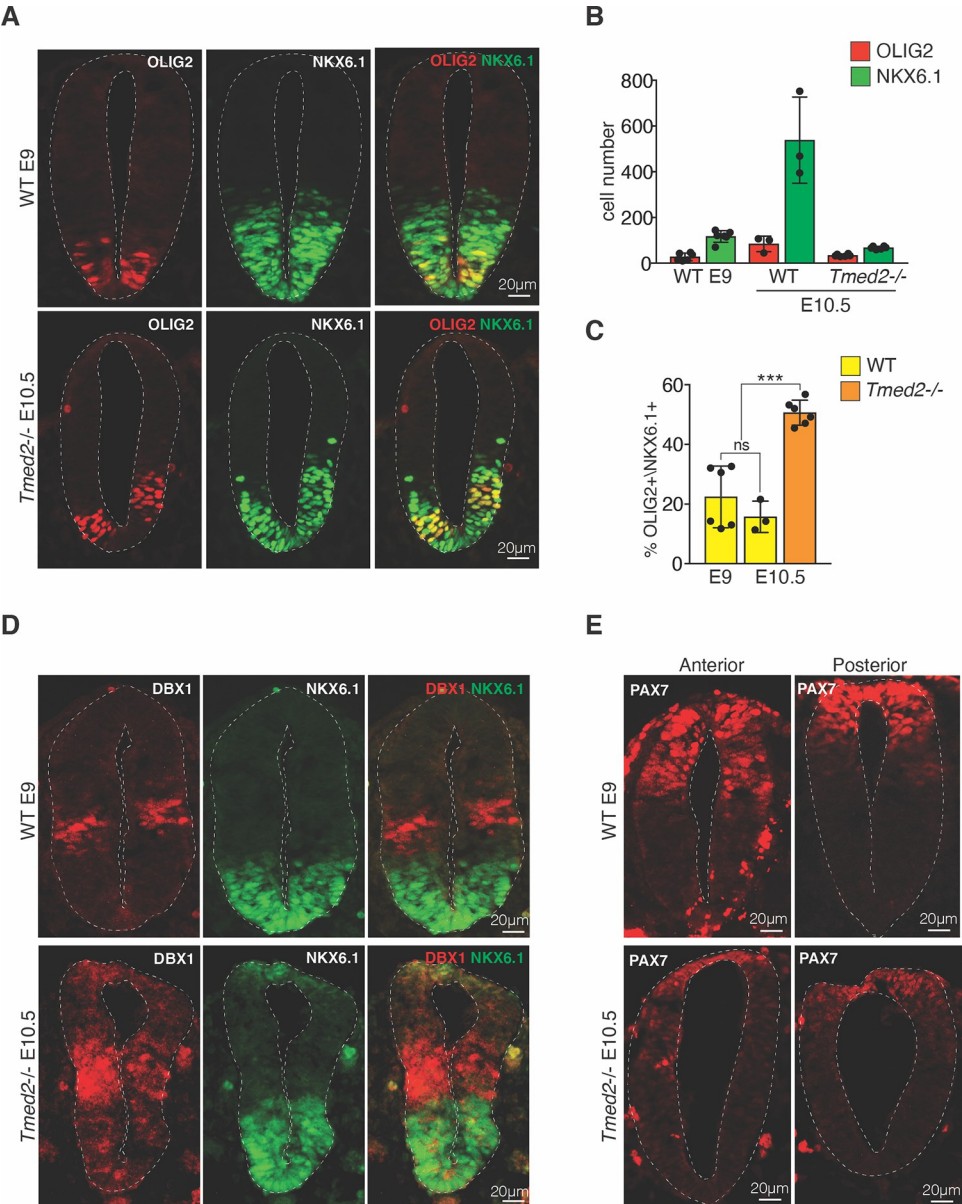

**Fig 5. Mutation of *Tmed2* affects neural tube pattering. (A–C)** Mutation of *Tmed2* increases expansion of the OLIG2 expression domain. (A) Neural tube sections of E9.0 control and E10.5 *Tmed2*$^{-/-}$ embryos were stained for the ventral markers OLIG2 and NKX6.1. Scale bar = 20 μm. (B) Quantification of OLIG2 and NKX6.1 expressing NPCs and (C) the percentage of NPCs expressing OLIG2 relative to NKX6.1 positive NPCs in neural tube section of E9.0 and E10.5 control, as well as E10.5 *Tmed2*$^{-/-}$ embryos. Asterisks denote statistical significance for difference between indicated samples. **(D)** Mutation of *Tmed2* promotes expansion of the DBX1 expression domain. Neural tube sections of E9.0 control and E10.5 *Tmed2*$^{-/-}$ embryos were stained for the markers DBX1 and NKX6.1. Scale bar = 20 μm. **(E)** *Tmed2*$^{-/-}$ embryos show decreased PAX7 expression. Neural tube sections derived from E9.0 control, and E10.5 *Tmed2*$^{-/-}$ embryos were stained for the dorsal marker PAX7. Scale bar = 20 μm. Anterior and posterior sections of the neural tube are shown for E9.0 contro, and E10.5 *Tmed2*$^{-/-}$ embryos. The data underlying all the graphs shown in the figure are included in the S1 Data file. NPC, neural progenitor cell; WT, wild-type.

and peripheral vesicular structures. Immunofluorescence experiments suggested overlap of the SMO staining with the ER marker ERP72 and with the LE marker RAB7. We also detected a cluster of SMO protein in a region located in front of the nucleus, which likely corresponds to

the microtubule organizing center (MTOC) involved in vesicle trafficking connected to the Golgi apparatus. Indeed, a fraction of SMO costained with the Golgi (RCAS1) and trans-Golgi network (SYNTAXIN-6) markers (S7B Fig). We confirmed these observations with an N-terminally tagged SMO (HA–SMO) (S7C and S7D Fig). We further assessed the localization of the constitutive active SMO-A1 mutant, which includes a W539L substitution. SMO-A1 accumulated in the ER (S7C Fig) consistent with an earlier observation [34], but no localization in the Golgi compartment was observed (S7D Fig). Considering that SMO-A1 was expressed at an estimated 20-fold higher level than our SMO–HA construct (S7E Fig) the Golgi localization of SMO–HA is unlikely a result of overexpression.

To further rule out potential effects from our SMO–HA expression construct, we investigated the intracellular distribution of SMO at the endogenous expression level. As we were unsuccessful of detecting SMO protein with antisera in the cell with the exception of the primary cilia, we introduced a 3xHA tag into the endogenous *Smo* locus using CRISPR nuclease-mediated templated repair (Fig 6H, S8A–S8C Fig). We were able to detect SMO–HA at the endogenous level in about 30% of the cells using a highly specific HA antiserum with a similar distribution as we had observed with our SMO–HA expression construct (S8D and S8E Fig). Importantly, endogenous SMO–HA and TMED2 colocalized in a restricted domain (Fig 6I). Although endogenous SMO–HA staining is weak and difficult to detect, it suggests that our observations with the SMO–HA expression construct are comparable and relevant for the physiological situation. These data strongly indicate that in SMO–HA WT cells, the SMO cluster colocalizing with the Golgi markers is specific and unlikely caused by artifacts of unfolding, tagging, or heterologous expression.

The resolution limits of conventional fluorescence microscopy considerably impaired our ability to observe colocalization with certainty. To obtain high-confidence data, we decided to perform dual-color super-resolution imaging. 3D-STORM analysis with adaptive optics allowed us to reach a resolution greater 50 nm laterally and 100 nm axially. We clearly detected localization of SMO–HA in the ER, Golgi, and LE compartments, but not in early endosomes (EEs) (Fig 6A–6C, S7F–S7I Fig). Performing coordinate-based colocalization (CBC) tests we verified the co-occurrence of SMO and these markers at a distance of less than 100 nm that correspond to the diameter of a vesicle.

TMED2 was more restricted than SMO and mainly detected in the Golgi compartment (Fig 6D). SMO–HA partially overlapped with TMED2 in specific and restricted domains (Fig 6D). Quantification showed that 40% of TMED2 is tightly associated with SMO (Fig 6E). To further explore an interaction between SMO and TMED2, we performed co-immunoprecipitation (co-IP) of SMO–HA in ESCs. co-IPs contained TMED2, showing that endogenous TMED2 can bind SMO–HA (Fig 6F). However, TMED10 was not detected in the co-IPs under our conditions, suggesting that TMED10 and SMO do not directly interact. We also confirmed a biochemical interaction of TMED2 and SMO in NPCs (Fig 6G). Taken together, our results identify TMED2 as a new binder of SMO and localize their interaction in the Golgi compartment.

## TMED2 is involved in SMO retention at the ER–Golgi compartments

Our findings suggested a role of the TMED2–SMO complex as an early step controlling HH signaling activation. In order to explore this possibility, we differentiated *Tmed2$^{-/-}$* ESCs expressing SMO–HA to NPCs and analyzed the effects on SMO distribution.

The *Tmed2* mutation did not affect the overall morphology of NPCs (S9A–S9D Fig). Small differences were observed in agreement with the literature [59,60] including accumulation of the ERP72 marker in localized regions of the ER (S9A Fig).

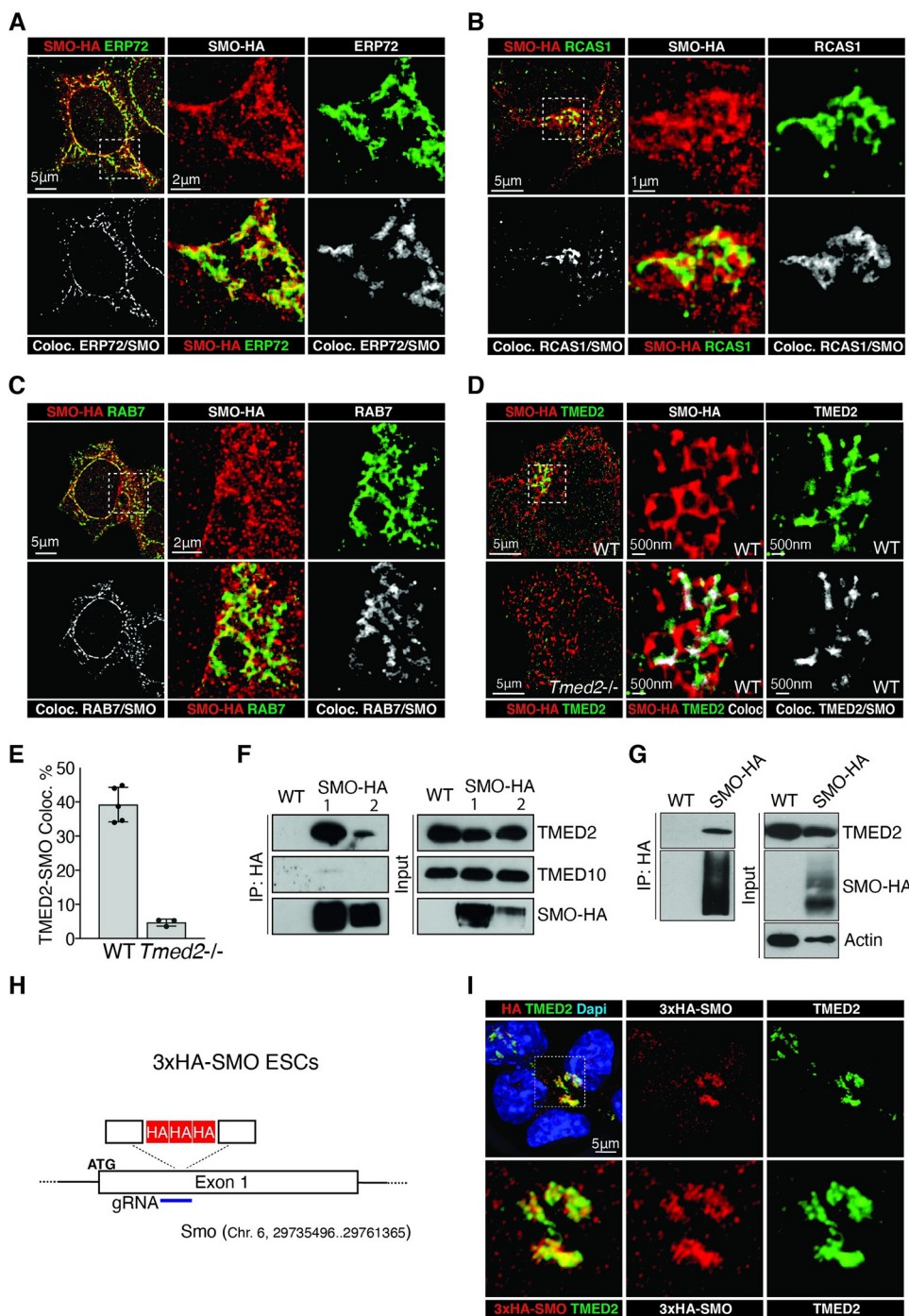

**Fig 6. SMO interacts with TMED2 in the Golgi compartment. (A–C)** Super-resolution SMO colocalization with (A) ERP72, (B) RCAS1, and (C) RAB7. SMO–HA cells were differentiated into NPCs and SMO–HA cellular distribution (red) was compared to organelle marker staining (green) in dual-color 3D-STORM experiments. Colocalizing events are shown in independent plots and labeled in gray; scale bar = 5 μm. Magnified areas are shown on the left; scale bar = 2 μm. **(D)** SMO colocalizes with TMED2 in NPCs in dual-color 3D-STORM experiments. Colocalization between SMO–HA (red) and TMED2 (green) is shown in the top left panel and compared to the one detected in parental cells with a *Tmed2* mutation (bottom right); scale bar = 5 μm. Magnified area is shown on the left; scale bar = 500 nm. Colocalizing events are labeled in gray. **(E)** Quantification of the colocalizing events in SMO–HA NPCs WT and depleted for *Tmed2*. **(F)** TMED2 binds SMO in ESCs. Western analysis of coimmunopurification of endogenous TMED2 and TMED10 with SMO–HA (left) and input (1/25 of IP, right) from extracts of 2 independent ESCs clones expressing SMO–HA. **(G)** TMED2 binds SMO in NPCs. Western analysis of coimmunopurification of

endogenous TMED2 with SMO–HA (left) and input (1/25 of IP, right). **(H, I)** Endogenous SMO colocalizes with TMED2 in NPCs. (H) Scheme showing the CRISPR/Cas-9 strategy used to endogenously tag the *Smo* gene with a 3xHA epitope in ESCs. (I) FM colocalization between endogenous 3xHA–SMO and TMED2 in experiments. Scale bar = 5 μm. Magnified area is shown on the bottom. The data underlying all the graphs shown in the figure are included in the S1 Data file. ESC, embryonic stem cell; FM, fluorescence microscopy; gRNA, guide RNA; HA, hemagglutinin; NPC, neural progenitor cell; SMO, Smoothened; WT, wild-type.

Treatment of NPCs with a low activating concentration of PMP led to translocation of SMO from internal to external compartments (Fig 7A). The perinuclear SMO staining detected in untreated NPCs diminished after chemical activation of SMO. Stimulation with SHH also promoted a similar SMO translocation but not to the extent of that induced by PMP. 3D-STORM experiments indicated that the mutation of *Tmed2* phenocopied the effects of SHH ligand on SMO localization (Fig 7B). Ciliary accumulation of SMO after SHH or PMP treatment was detected in few cells. This is likely explained by cell proliferation in NPC cultures that prevents the formation of clear cilia. Next, we analyzed the effects of the *Tmed2* mutation on SMO distribution performing dual color 3D-STORM imaging. We observed a consistent reduction of the area in the ER and LE that was occupied by SMO in $Tmed2^{-/-}$ cells compared to control cells (Fig 7C, 7E, and 7F, S10A–S10D Fig). T reduction was even more pronounced in the Golgi (Fig 7D, S10C Fig). In *Tmed2* mutant cells, colocalization of SMO with RCAS1 was 3-fold lower thhean in control cells (Fig 7F). Notably, treatment with SHH induced a comparable relocalization of SMO as the *Tmed2* mutation. SHH treatment reduces the SMO pool in the ER, LE, and Golgi compartments. Besides these similarities, a difference also emerged. In *Tmed2* mutant cells, we detected a global decrease of SMO in the ER, whereas in cells treated with SHH, SMO depletion was more pronounced in ER peripheral domains (Fig 7C, S10B Fig). Perinuclear ER regions are enriched in cisternal domains associated with ribosomes and polysomes and therefore involved in protein synthesis [69]. We believe that the detected difference reflects the short-term effect of the SHH treatment, compared to a constitutive loss of *Tmed2*. According to this interpretation, SHH treatment first mobilizes the peripheral pool of SMO.

## TMED2–SMO complex is regulated by SHH and modulates SMO abundance at the plasma membrane

To explore SMO trafficking after its release form the ER and Golgi compartments, we analyzed SMO colocalization with the EE marker EEA1. EEs are strongly interconnected with the PM and involved in recycling GPCRs from the cell surface. However, treatment with SHH or depletion of *Tmed2* did not lead to an increase of the SMO fraction in EEs (S11A Fig, Fig 7F). We analyzed SMO translocation to the plasma membrane by using a cell-impermeable biotinylation reagent and purification of biotinylated proteins. Consistent with immunofluorescence experiments, a small amount of SMO–HA was detected at the plasma membrane in control cells in the absence of SHH ligand. SHH treatment–induced SMO–HA translocation to the plasma membrane (Fig 7G, S11B Fig). In *Tmed2* mutant cells, SMO–HA abundance was dramatically increased at the plasma membrane to a level seen in control cells after SHH treatment. The TMED2 mutation had no effect on the total cellular amount of SMO–HA. Furthermore, epidermal growth factor receptor (EGFR; Fig 7G) and N-Cadherin (S11C Fig) secretion were not perturbed in mutant cells showing that the *Tmed2* mutation had a specific effect on SMO translocation but did not cause a general secretory defect. We further investigated the GPCR Fzd2, which is closely related to SMO. Fzd2 can bind TMED2 in a biochemical pull-down experiment, but its plasma membrane localization is not elevated by the *Tmed2* mutation (S6D and S6E Fig).

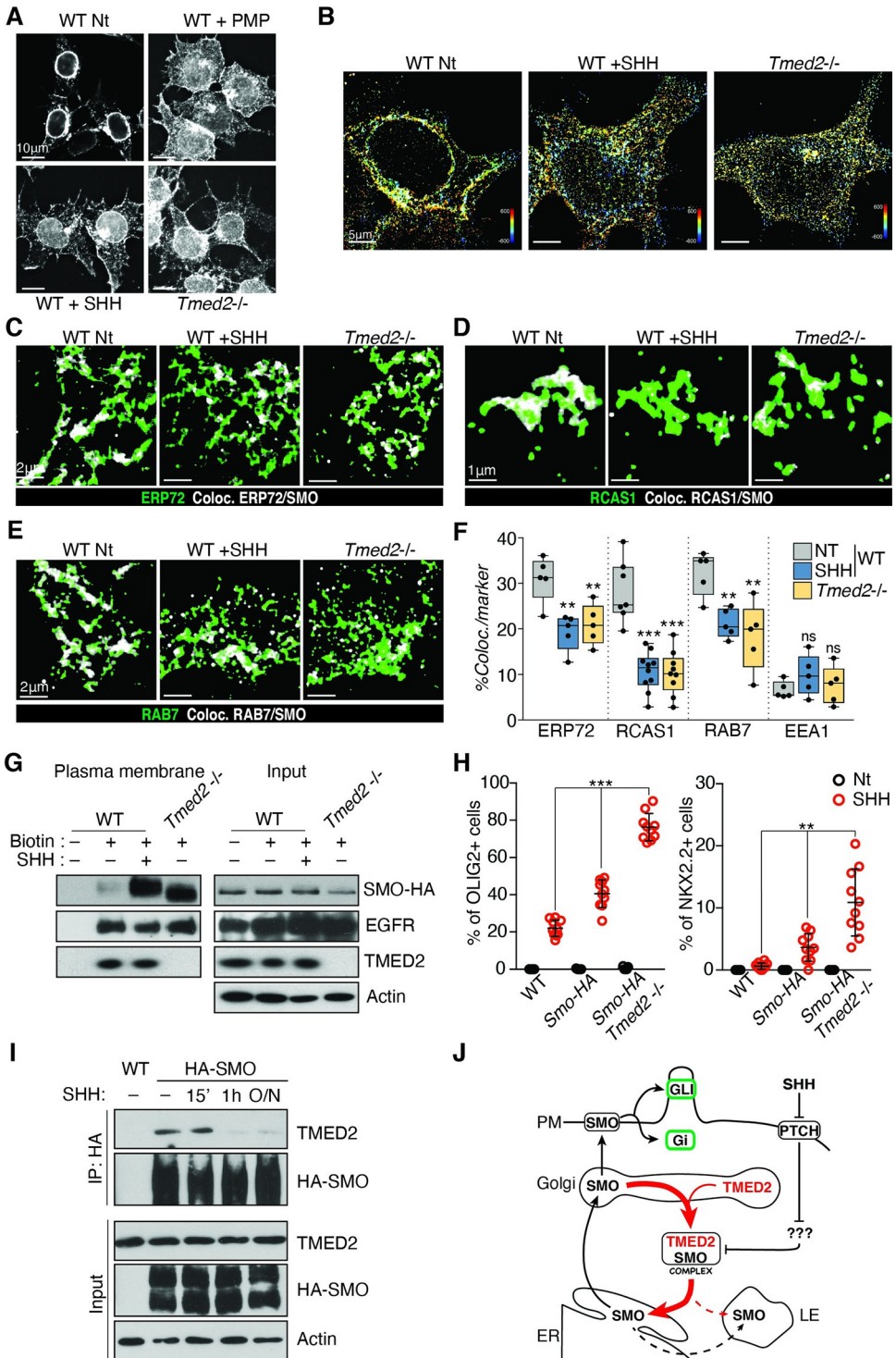

**Fig 7. TMED2 regulates SMO abundance at the PM in a SHH-dependent manner. (A, B)** Activation of HH signaling promotes SMO cellular redistribution. (A) FM images of SMO–HA in WT NPCs treated for 24 hours with PMP or SHH and in *Tmed2*$^{-/-}$ cells. Scale bar = 10 μm. (B) Three-dimensional color distribution of HA–SMO in 3D-STORM experiments performed in WT NPCs treated with SHH for 24 hours and in *Tmed2*$^{-/-}$ NPCs. Scale bar = 5 μm. **(C–F)** SHH treatment and *Tmed2* mutation promotes SMO trafficking form ER–Golgi compartments. Dual color 3D-STORM analysis showing ERP72 (C), RCAS1 (D) and RAB7 (E) distribution in green and the SMO colocalizing domains in gray. (F) Plot showing percentage of SMO–HA colocalizing events normalized to marker distribution upon SHH treatment and *Tmed2* mutation. Asterisks denote statistical significance for difference from the

WT untreated samples. **(G)** *Tmed2* regulates SMO abundance at the PM. Cells were treated with NHS-SS-Biotin (Biotin) to label PM proteins, and with SHH as indicated. Western analysis of PM proteins and input before purification (1/50 of pull-down) are shown. **(H)** Percentage of cells expressing OLIG2 (left) and NKX2.2 (right) relative to total cell count in neuralized EBs. Neuralized EBs derived from WT cells and from ESCs overexpressing SMO–HA with and without a *Tmed2* mutation were treated with or without SHH. Asterisks denote statistical significance for difference between indicated samples. Individual EBs are plotted (*n* = 10). **(I)** SHH treatment disrupts the SMO–TMED2 complex in NPCs. Western analysis of co-immunopurification of endogenous TMED2 with HA–SMO (top) and input (1/25 of IP, bottom) in NPCs expressing N-term HA tagged SMO. Cells were treated with recombinant SHH for the indicated amount of time. Actin is shown as loading control. **(J)** Summary of the proposed mechanism for TMED2-regulated SMO secretion from the ER–Golgi compartment. The data underlying all the graphs shown in the figure are included in the S1 Data file. EB, embryoid body; EGFR, epidermal growth factor receptor; ER, endoplasmic reticulum; ESC, embryonic stem cell; FM, fluorescence microscopy; HA, hemagglutinin; HH, hedgehog; LE, late endosome; NPC, neural progenitor cell; NT, not treated; PM, plasma membrane; PMP, purmorphamine; SHH, Sonic hedgehog; SMO, Smoothened; WT, wild-type.

Our data demonstrate that TMED2 specifically regulates the abundance of SMO at the PM. The cytoplasmic dot pattern detected by immunostaining in SHH-treated and *Tmed2* mutant NPCs likely correspond to SMO trafficking through the PM.

To assess if SMO–HA from our expression construct would have a synergistic effect with the *Tmed2* mutation for ectopic activation of Gli transcription, we analyzed the ventral markers OLIG2 and NKX2.2 in neuralized embryoid bodies (EBs). We find that the number of cells expressing ventral markers in the presence of exogenous SHH increases from wild-type, SMO–HA transgenic, to SMO–HA transgenic *Tmed2* mutant EBs (Fig 7H). However, ventral markers were not activated in the absence of SHH in any of the genotypes, showing that transgenic SMO–HA does not cause ectopic GLI activation in the absence of HH ligands consistent with our earlier experiments with endogenous SMO (Fig 4F). This observation further suggests that enhanced HH signaling in *Tmed2* mutant embryos is dependent on HH ligand and a *Tmed2* mutation by itself does not lead to ectopic GLI activation. The increased abundance of SMO at the PM in the absence of *Tmed2* renders the cells hypersensitive to SHH.

The similarity between SMO release in *Tmed2* mutant and SHH-treated cells prompted us to investigate a role for SHH in modulating the binding of TMED2 to SMO. We performed HA co-IPs in NPCs derived from HA–SMO expressing ESCs. In the absence of SHH ligand, TMED2 co-immunoprecipitated with HA–SMO in NPCs (Fig 7I), consistent with our previous results. Addition of SHH led to a loss of the interaction between SMO and TMED2 (Fig 7I). The interaction was lost as early as 1 hour after SHH treatment and remained at low levels in the presence of SHH. The interaction between SMO and TMED2 was confirmed with a carboxyl-terminally tagged SMO–HA construct (S11F Fig). Also, in this case, SHH treatment disrupted the SMO–TMED2 complex with similar kinetics. Although the mechanism of how the HH signal is transduced from PTCH to the Golgi–ER boundary remains unknown, the resolution of the TMED2–SMO interaction is one of the earliest effects after signal reception.

## Discussion

Our study identified a new function of the p24 family members TMED2 and TMED10 in HH signaling. We show that TMED2 acts as a novel repressor of HH signals in development by regulating SMO levels at the plasma membrane. Selection of *Tmed10* in our screen is explained by its function in maintaining TMED2 protein in ESCs. *Tmed2* acts upstream of and affects both the GLI-dependent and GLI-independent effects of SMO. The mutation of *Tmed2* leads to higher abundance of SMO at the plasma membrane, but does not by itself allow SMO to enter the ciliary compartment or to induce GLI processing in the absence of a HH signal. Therefore, no ectopic activation of basal GLI activity is observed. The higher abundance of

SMO at the plasma membrane increases the strength of the GLI response after a HH signal is received.

Our finding raises the question of how TMED2 regulates SMO abundance at the plasma membrane. The majority of TMED2 resides in the Golgi, and in 3D-STORM experiments, we detected an overlap between TMED2 and SMO with a resolution of less than 50 nm. We show that TMED2 and SMO also interact in biochemical experiments. Based on the implication of TMED2 in vesicle trafficking [59,65–68], our data lead us to propose a model for TMED2 function in the retrograde transport of SMO between the Golgi and the ER compartment (Fig 7J). In our model, SMO traffics between the ER and Golgi compartments and does not reach the plasma membrane in the absence of HH signaling. In *Tmed2* mutant cells, SMO is no longer retained and free to traffic to the cell surface.

Our observation of an interaction between SMO and TMED2 is consistent with reports of interaction of TMED2 with other GPCRs. TMED2 arrests PAR-2, P2Y4 receptor, and l-opioid receptor 1B at the intracellular compartments [66]. In contrast, increased plasma membrane abundance of the calcium sensing receptor (CaSR) has been described after TMED2 overexpression [68]. We also have obtained evidence for an interaction between the Wnt receptor FZD2 and TMED2. In our experiment, the *Tmed2* mutation did not affect the abundance of FZD2 on the plasma membrane. These findings support a general role of TMED2 in GPCR trafficking. However, the functional consequences of TMED2 binding are different and dependent on the specific GPCR. The complexity of defects in *Tmed2* mutant embryos supports this view [62]. Our data suggest that misregulated HH signaling is one component of the *Tmed2* mutant phenotype.

Importantly, we demonstrate that the interaction between SMO and TMED2 is disrupted by HH signals in NPCs. This finding identifies a new regulatory function of TMED2 in HH signaling and development. A ligand-mediated regulation of TMED2 interaction has not been proposed for other GPCRs previously. One of the most evident effects of SHH treatment is the unloading of SMO from the Golgi apparatus. The observation that SHH or chemical agonists promote the release of SMO to reach the plasma membrane is consistent with earlier reports of a release of SMO from internal compartments [12,51]. Our super-resolution imaging shows that HH signals induce a similar relocalization of SMO from the internal compartments to more peripheral compartments as the mutation of *Tmed2*. The Golgi compartment is crucial for unraveling the mechanism of HH pathway activation. We believe that the high proportion of Golgi related genes detected in our screening reflect this relevance. Screening for factors promoting SMO levels and activation, we are investigating the earliest steps of the HH cascade. This aspect suggests similarity in the upstream regulation between the mammalian and invertebrate HH pathway. While in *Drosophila* translocation of SMO to the PM represents an activating event [35], in mammals, SMO has been considered to constantly traffic between the PM and the recycling compartment [70]. In our analysis, SMO weakly localized in the EEs vesicles, and this pool appears not to be affected by SHH stimulation. Our observations suggest that SMO abundance at the PM is directly regulated by HH signals and depends on the release from the ER–Golgi compartments.

At the moment, the molecular nature of the signal that leads to the loss of the SMO TMED2 interaction is unknown. We observe a higher molecular weight form of SMO on the plasma membrane after HH signaling activation. In *Tmed2* mutant NPCs, SMO reaches the plasma membrane with a lower molecular weight. The molecular weight differences are likely explained by different glycosylation of the protein. This could suggest glycosylation of SMO as a signal for release from TMED2 binding. However, we consider this an unlikely possibility. First, in ESCs, the glycosylated form of SMO reaches the plasma membrane in the absence of *Tmed2*. Therefore, the observed differences in glycosylation could reflect either the time SMO

requires to pass through the Golgi or selection of an alternative route bypassing the Golgi [38]. An earlier study has associated the glycosylation of SMO with GPCR activity [52]. This is consistent with our observation that SMO GPCR function rescues cell death in ESCs. In contrast, in *Tmed2* mutant NPCs the nonglycosylated form of SMO accumulates at the plasma membrane and upon receiving a HH signal is capable of inducing an elevated GLI response. The observation that a HH signal is required for a GLI response shows that GLI activation is regulated at 2 steps. First, SMO translocation to the plasma membrane is regulated by TMED2. Secondly, entry into the ciliary compartment is regulated by an independent mechanism. Our data show that the *Tmed2* mutation allows to mechanistically separate the trafficking of SMO to the plasma membrane from its ciliary localization. Formally, we cannot exclude other mechanisms of the resolution of the SMO and TMED2 interaction. It is conceivable that routing of SMO changes after a HH signal has been received and SMO might bypass TMED2 in the Golgi. Considering the literature on membrane protein trafficking and the fact that an additional internal regulated step for SMO trafficking would need to exist, we deem this model unlikely at the current time.

The finding that SHH disrupts TMED2–SMO interaction raises the question of how SHH signaling could be transduced to the ER–Golgi compartment. The signal can be expected to originate from the receptor PTCH1. PTCH1 continuously traffics through the plasma membrane and LE compartment and after SHH binding is subjected to degradation [6]. Our superresolution imaging indicates that before activation, a relevant fraction of SMO also localizes in the LE compartment. It is conceivable that LEs might represent a meeting place where PTCH1 is in close proximity with SMO. In this confined compartment, PTCH1 can promote modifications preventing SMO maturation and secretion. Even if at the moment speculative, this hypothesis has some support from previous work trying to characterize SMO and PTCH1 subcellular trafficking [34,71]. Our study provides a novel entry point for unraveling the enigmatic signal transduction mechanism from the receptor PTCH to the transducer SMO.

## Materials and methods

### Cell culture

Mouse ES cells were cultured in chemically defined 2i medium plus leukemia inhibitory factor (LIF) as described with minor modifications [39,72]. The 2i medium was supplemented with nonessential amino acids and 0.35% BSA fraction V. Culture of ES cells on feeders was performed as previously described [73]. NIH-3T3 and 293T cells were obtained from ATCC and grown in DMEM +10% fetal bovine serum (FBS) media supplemented with antibiotics. $Ptch1^{-/-}$ MEFs were provided by M. P. Scott, Stanford University School of Medicine.

**Derivation of haploid ESCs.** Haploid ESCs from 129S6/SvEvTac Oocytes (ha129DM1) were derived as previously described [74]. Briefly, oocytes were isolated from superovulated female mice and activated in KSOM medium using 5 mM strontium chloride and 2 mM EGTA. Embryos were subsequently cultured in Cleavage (Cook Medical (Cook Medical Inc., Indiana, USA), G20720) medium microdrops covered by mineral oil. After 4 to 5 days, embryos at the morula or blastocyst stage were treated with acidic Tyrode's solution to remove the zona pellucida, and ESCs derivation was performed as previously described.

### Mice and embryo section

All animal procedures were approved by the veterinary office of the Canton of Zurich and the Canadian Council on Animal Research animal (license numbers: 29340; ZH152/2017). The $Tmed2^{99/99J}$ ($^{-/-}$) mouse line was described previously [62] and genotyped by PCR using the following primers: *Tmed2*In4F (AAGTGCACAGCTGAGTGGT) and *Tmed2*In4R

(CACAGTGTCTGACCCCCTTT). Embryos of E10.5 *Tmed2*$^{-/-}$ mice were compared to E9.0 WT embryos (CD1 strain). For sections, embryos were fixed with 4% paraformaldehyde (w/v), cryoprotected with 30% sucrose (w/v) overnight, and embedded in OCT (Leica, Germany). Moreover, 10-μm horizontal cryotome sections (E9.0, E9.5, and E10.5) were processed for immunofluorescence staining.

## Reagents

**Growth factors and chemicals.** The following growth factors and chemicals were used: LIF (recombinant purified as GST fusion protein, homemade), mouse sonic hedgehog C24II (SHH, recombinant purified as GST fusion protein), mouse epidermal growth factor (EGF, PeproTech), human basic fibroblast growth factor (bFGF, PeproTech), CHIR99021 (Axon Medchem, Groningen, The Netherlands), PD0325901 (Axon Medchem), SAG (Calbiochem, Sigma-Aldrich, Missouri, USA), PMP (Calbiochem), Cyclopamine-KAAD (Calbiochem), SANT-1 (Sigma, Sigma-Aldrich, Missouri, USA), GANT61 (Sigma), and Nocodazole (Sigma).

**SHH production.** The pcDNA3-Shh (N) (Addgene (Massachusetts, USA), #37680) was used as template to amplify the cDNA coding residues 24–197 of mouse *Shh*. By PCR, a Cys24-Ile-Ile substitution and a Factor Xa cleavage site were introduced. The construct was cloned in the pGEX vector and transformed in BL21 (DE3) pLysS *Escherichia coli* cells. The transformed cultures were grown in a shaking incubator at 37˚C until OD600 of 0.8 was reached, at which point the temperature was switched to 30˚C, and the expression was induced with 0.5 mM IPTG. After 4 hours, cells were collected by centrifugation, frozen, and stored at −80˚C until later use. ShhNC24II-expressing *E. coli* pellets (0.1 l culture) were thawed, resuspended in 5 ml of lysis buffer (25 mM sodium phosphate (pH 8.0), 150 mM NaCl, 1 mM EDTA, 0.5 mM DTT and 1 mM PMSF), disrupted by sonication, and cleared by centrifugation for 30 minutes at 25,000 × g. The supernatant was added to Glutathione Sepharose 4B Resin (Sigma, GE17-0756-01) and incubated O/N at 4˚C. The resin was collected and washed 5 times in Lysis Buffer. The protein was eluted by digestion with Factor Xa protease (Sigma). Factor Xa protease was removed using p-Aminobenzamidine-agarose (Sigma) and recombinant SHH was purified with Detoxi-Gel Endotoxin Removing Gel (Thermo Fisher Scientific, Massachusetts, USA) and dialyzed. SHH was supplemented with 10% glycerol, aliquoted into 100 μl batches, flash-frozen in liquid nitrogen, and stored at −80˚C.

**Antibodies.** The following antibodies were used: αTMED2 (Santa Cruz Biotechnology (Texas, USA), sc-376459), αTMED10 (sc-137003), αSMO (sc-166685), αActin (Sigma A5316), αHA (Roche (Basel, Switzerland), 12013819001) (Cell Signaling Technology (Massachusetts, USA), 3724) (BioLegend (California, USA), #901501), αARL13B (Proteintech (Illinois, USA), 17711-1-AP), αRCAS1 (Cell Signaling Technology #12290), αERp72 (Cell Signaling Technology #5033), Rab7 (Cell Signaling Technology #9367), Syntaxin-6 (Cell Signaling Technology #2869), EEA1 (Cell Signaling Technology #3288), αOLIG2 (Merck-Millipore (Massachusetts, USA), AB9610), αNKX2.2 (DSHB, University of Iowa, USA), αNKX6.1 (DSHB), αPAX7 (DSHB), αPAX6 (Novus (Novus Biologicals, Colorado, USA), NBP195459), HSP-90 (sc-13119), E-cad (BioLegend, #866701), N-cad (BioLegend, #844702), EGFR (Cell Signaling Technology #4267), Dbx1 [75], and Phospho-Myosin Light Chain 2 (Thr18/Ser19) (Cell Signaling Technology #3674).

## Cell treatment

**Flow cytometry and activated cell sorting.** For derivation and maintenance of haploid ESCs, cell sorting for DNA content was performed after staining with 15 mg/ml Hoechst

33342 (Invitrogen, Massachusetts, USA) on a MoFlo flow sorter (Beckman Coulter, California, USA) selecting for the haploid 1n (G1) peak.

**Transfection and stable cell line generation.** For generation of $Tmed2^{-/-}$, $Tmed10^{-/-}$, and $Smo^{-/-}$ ESCs, the following guide RNAs (gRNAs) sequences were cloned in the pX458_pSpCas-9(BB)-2A-GFP vector (Addgene, #48138):

gRNA*Tmed2*_MscI GCTGGCCGCGCTGCTGGCCA
gRNA*Tmed10*_FspI GCACCTTGAGGTGGGTGCGC
gRNA*Smo*_BsrBI CCCCTGTGCCATTGTGGAGC
gRNA*Smo*_XhoI GCGCCAGCGGGAGCTCGAGG

Briefly, plasmid vectors were used for transfection (Lipofecatmine 2000) of haploid 129DM1. Forty-eight hours later, cells were sorted for green fluorescence and plated at low density for isolating individual clones. Mutations were identified by PCR on genomic DNA and sequencing using the following primers:

*Tmed2* Fwd: CTCCGGAGGCCGCAGT
Rev: GACGAGCGCTTTCCGAGA
*Tmed10* Fwd: TCTGGTTTGTTTGGCCCACTCT
Rev: AAGTGGAAACAGCCCTAGGTCTC)
*Smo_1st transcript* Fwd: TTTGCTGAGTTGGCTGTTTG
Rev: TACTCGGGCTCTTTGTGACC
*Smo_2nd transcript* Fwd: GGAGGGGTCTTTGCCACGAT
Rev: ACGGGGAAGGAAAAAGAAAA

For derivation of HA–SMO and SMO–HA ESCs mouse, *Smo* was amplified by PCR from the pGEN-*mSmo* (Addgene, #37673) and introduced in the PB-HA-IRES-Neo vector. Stable integration of the construct was obtained in $Smo^{-/-}$ ESCs by cotransfection of the Piggybac and the PBase plasmids. Cells were plated by limiting dilution and integration events selected with G418. Two clones characterized by high and low HA–SMO or SMO–HA expression were expanded. SMO–HA $Tmed2^{-/-}$ ESCs were derived from SMO–HA ESCs with low SMO–HA expression as previously described.

For derivation of *Fzd2*-HA ESCs mouse, *Fzd2* was amplified by PCR from the pRK5-*mFzd2* (Addgene, #42254) and introduced in the PB-HA-IRES-Neo vector. Stably expression of the construct was achieved in WT and $Tmed2^{-/-}$ ESCs as previously described.

Transfections of NIH-3T3 cells with siRNA targeting mouse *Tmed2* (Fwd: CACCU-CUAAUUGAAUUGAACAAGCA, Rev: UGCUUGUUCAAUUCAAUUAGAGGUGAU) were performed with RNAiMax (Invitrogen). The Negative Control DsiRNA (IDT (Integrated DNA Technologies (IDT), Iowa, USA), 73481795) was used as transfection control.

**Genomic tagging of SMO using CRISPR/Cas-9–mediated homologous recombination.** The sequence coding for the 3-HA epitope was inserted at position 153 of mouse *Smo* cDNA (exon 1), as in the PB-HA–SMO-IRES-Neo construct previously described. Cas-9 recombinant protein was kindly provided by the Jinek Lab [76].The oligonucleotide used for HR process and the gRNA targeting the SMO locus (gRNA*Smo*_XhoI) were ordered from IDT.

SMO_HR_Cas-9_EcoRI_3xHA-Nterm: tgagcgggaacgtgaccgggcctgggcctcacagcgccagcgg-gagctcgGAATTCTACCCATACGATGTTCCAGATTACGCTGGCTACCCATACGATG TTCCAGATTACGCTGGCTACCCATACGATGTTCCAGATTACGCTaggaggAacgtgccggt-gaccagccctccgccgccgctgctgagccactg gRNA*Smo*_XhoI:

mGmCmGrCrCrCArGrCrGrGrGrArGrCrUrCrGrArGrGrGrGrUrUrUrUrArGrArGrCrUrAr-GrArArArUrArGrCrArArGrUrUrArArArArUrArArGrGrCrUrArGrUrCrCrGrUrUrArUr-CrArArArCrUrUrGrArArArArArArGrUrGrGrCrArCrCrGrArGrUrCrGrGrUrGrCmUmUmUrU

For transfection of the Cas-9 RNP complexes into ESCs cells, a NEPA21 electroporator (Sonidel (Sonidel Limited, Dublin, Ireland)) was used following settings described in [76]. Electroporated cells were plated by limiting dilution and integration events were screened by PCR using the following primers:

*Smo*-ext_FW TTGCAAAGTTGGGAGTCGAGG

*Smo*-ext_RV CAGAGTCTCCTTCCCGCAC

SMO–HA_FW AGCGGGAGCTCGGAATTCTAC

**Drug treatment and survival assay.** ESCs and NIH-3T3 were treated with SHH (50 to 500 ng/ml), SAG (0.5 to 5 μM), PMP (1 to 10 μM), Cyclopamine-KAAD (1 to 5 μM), or SANT-1 (10 to 50 μM). After 48 hours, cells were trypsinized and counted. Survival is expressed normalizing cell counts to DMSO treated sample. To evaluate effects of compounds on GLI transcriptional activity, cells were treated with SHH (500 ng/ml), SAG (5 μM), PMP (10 μM), Cyclopamine-KAAD (5 μM), or SANT-1 (50 μM).

**Hedgehog signaling assays.** NIH-3T3 cells were grown to confluency in DMEM containing 10% FBS. Confluent cells were cultured in 0.5% FBS DMEM for 24 hours to allow ciliogenesis prior to treatment with drugs and/or ligands in DMEM containing 0.5% FBS for various times, as indicated in the figures.

**NPCs derivation from ESCs.** Differentiation of ESCs to NPCs in 2D conditions was previously described [77]. Briefly, the cells were plated on Matrigel (Matrigel hESC-Qualified Matrix, LDEV-free (Corning, New York, USA)) in N2B27 media (DMEM- F12 Gibco (Gibco, Thermo Fisher Scientific, Massachusetts, USA) and Neurobasal Medium (Gibco) mixed in 1:1 ratio and supplemented with N-2 supplement, B-27 supplement, 1% penicillin/streptomycin, 2 mM L-glutamine, 40 mg/ml Bovine Serum Albumin, and 100 μM 2-mercaptoethanol). On day 0 and day 1, cells were cultured in N2B27 with 10 ng/ml bFGF. On day 2, the media was changed and the cells were cultured in N2B27 with 10 ng/ml bFGF and 5 μM CHIR9902. On day 3, the media was changed, and the cells were cultured in N2B27 supplemented with retinoic acid (RA, 100 nM), and SHH (1–4 μg/ml). On day 4, an equal volume of N2B27 with 100 nM RA was added to each well diluting treatment condition in half. On day 5, cells were processed for further analysis. For derivation of neuralized EBs, ESCs were plated in EBs media (DMEM-F12 (Gibco) and Neurobasal Medium (Gibco) (1:1 ratio) supplemented with 200 mM L-glutamine and 10% of knockout serum replacement) on Sphericalplate 5D dishes (Kugelmeiers (Kugelmeiers Ag, Erlenbach, Switzerland)). On day 2, EBs were transferred in 10-cm tissue culture dishes and treated with RA (100 nM). EBs were treated with SHH (4 μg/ml) the day after, and neuralized EBs were collected and processed for analysis at day 5.

## Genome-wide screening

Haploid ESCs were mutagenized using a lentivirus system with a gene-trap cassette [78]. The pRRLsin-PPT-SA-PuroGFP-Wpre-pA plasmid was transfected with lentiviral packaging plasmids in 293T. Virus was collected and concentrated by ultracentrifugation to obtain a high viral titer. Moreover, $6 \times 10^7$ sorted haploid ESCs were infected with virus and plated on 145 cm$^2$ dishes precoated with MEFs. After 2 days, cells were collected and split into control and selected sample. Control cells were directly lysed, and DNA was extracted. For selection, PMP was added to the media at a concentration of 10 μM. Every 2 days, cells were passaged until day 14. Non infected cells were grown in parallel and treated like infected cells. To identify genomic insertion sites of genetrap viruses, NGS libraries were prepared and sequenced on an Illumina MiSeq device as described [79]. In brief, genomic sequences adjacent to the end of the viral LTR were enriched by linear amplification PCR (LAMPCR) using a high-fidelity DNA polymerase (Invitrogen, #12346094) with a biotinylated primer targeting the viral

genome. Single-stranded LAM PCR products were captured on Streptavidin coated magnetic beads (Dynabeads M270, Invitrogen #65305). Subsequently, an oligonucleotide containing the Illumina P7 adaptor sequence was ligated to the 3′ end of the single-stranded LAM PCR fragments using Circligase (Epicentre, California, USA/Illumina, California, USA, #CL9025K). A P5 adaptor was added by PCR. The resulting DNA was purified and concentrated using the MinElute PCR Purification Kit (QIAGEN (Hilden, Germany), #28004) before loading it on the Illumina MiSeq flow cell. Sequencing was performed using 75 cycles, pair-end runs on an Illumina MiSeq sequencer using MiSeq v3 kits (Illumina, #MS1023001). Computational analysis of NGS data sets was performed using the HaSAPPy package [56].

### RNA isolation and qPCR

Total RNA was isolated from ESCs using the QiAshredder (QIAGEN, #79656) and purified using RNeasy Mini Kit (QIAGEN, #74104) and on-column DNase I digestion (QIAGEN, #79254). cDNA for real-time PCR was synthesized using the QuantiTect Reverse Transcription Kit (QIAGEN, #205313). Real-time quantitative PCR reactions were performed using SYBR Green and a LightCycler 480 System (Roche). Relative expression of the target gene was normalized to *Eif4a2* and *Sdha* expression levels. Primer sequences:

*Gli1* Fwd: GAATTCGTGTGCCATTGGGG
Rev: GGACTTCCGACAGCCTTCAA
*Ptch1* Fwd: TGACTGGGAAACTGGGAGGA
Rev: TGATGCCATCTGCGTCTACC
*Eif4a2* Fwd: ACACCATCGGGGTCCATTCC
Rev: CCTGTCTTTTCAGTCGGGCG)
*Sdha* Fwd: TTCCGTGTGGGGAGTGTATTGC
Rev: AGGTCTGTGTTCCAAACCATTCC

### Protein extraction and immunoblotting

Whole cell extracts from NIH-3T3, ES cells, or NPCs were prepared in RIPA lysis buffer (50 mM Tris-HCl pH-7.4, 150 mM NaCl, 2% NP-40, 0.25% Deoxycholate, 0.1% SDS, 1 mM DTT, 10% glycerol, protease inhibitors). Samples were resuspended in Laemmli sample buffer, denatured, and subjected to SDS-PAGE. The resolved proteins were transferred onto a nitrocellulose membrane (Bio-Rad (Bio-Rad Laboratories Inc., California, USA)) using a wet electroblotting system (Bio-Rad) followed by immunoblotting.

**Co-immunoprecipitation experiments.** For co-IP experiments, cells were lysed in the Membrane lysis buffer (50 mM Tris-HCl pH 7.5, 1 mM β-mercaptoethanol, 150 mM NaCl, 1% ChAPS and proteases inhibitors) and quantified. An equal amount of proteins was incubated with anti-HA affinity matrix (Sigma, #11815016001) to pull-down SMO–HA. After 1 hour, samples were washed, denatured in Laemmli sample buffer and urea, and subjected to SDS-PAGE.

**Plasma membrane protein purification.** Biotinylation of cell surface proteins was performed as described previously [80]. Briefly, ESCs were differentiated to NPCs as previously described in 10-cm dishes. Cells were incubated for 30 minutes with Sulfo-NHS-SS-Biotin (CovaChem (Illinois, USA), #14207) on ice and lysed in membrane lysis buffer. Lysates were quantified, and an equal amount of protein lysate was incubated on Pierce High Capacity Streptavidin Agarose (Pierce (Thermo Fisher Scientific, Massachusetts, USA, #20357), # 20357) for 1 hour. Samples were washed, denatured in Laemmli sample buffer and urea, and subjected to SDS-PAGE.

## Immunofluorescence and localization studies

Cells were fixed in 4% PFA for 15 minutes at 37˚C and washed in PBS. Cells were incubated in blocking buffer (10% donkey serum, 0.3% triton, 1x PBS) for 30 minutes. Primary antibodies diluted in antibody buffer (1% BSA, 0.3% triton, 1x PBS) were added for 1 hour or O/N. Secondary antibodies and DAPI were added for 45 minutes in antibody buffer. To improve detection of TMED2 protein, cells were incubated in Golgi buffer (0.1% SDS, 10% 2-Mercaptoethanol, 10% donkey serum in PBS) for 30 minutes at 60˚C before incubation with the primary antibody. Samples were viewed on a Zeiss Image Z1 microscope equipped with an X-Cite 120 illuminator (EXFO) and a Leica SP8 confocal microscope.

For immunofluorescence staining of embryo sections, the sections were subjected to the following antigen retrieval procedure to increase signaling. Samples were incubated in 0.3% $H_2O_2$ diluted in PBS for 30 minutes. Antigens were recovered at 105˚C with a pressure cooker device for 15 minutes in prewarmed sodium citrate solution (1.8 mM citric acid monohydrate, 8.2 mM sodium citrate tribasic dihydrate). Sections were cooled down to room temperature for 2 hours and then incubated in 2N HCl for 3 minutes at 37˚C. Sections were incubated in TNB blocking solution (0.1 M Tris-HCl pH 7.5, 0.15 M NaCl, 0.5% FP1012 Blocking reagent (Perkin Elmer, Schwerzenbach, Switzerland)) at room temperature for 30 minutes. Primary and secondary staining were performed as previously described.

## STORM imaging

Images were collected on Nikon Ti2 with Perfect Focus System with a sCMOS camera (Hamamatsu Orca Flash 4 v3 (Hamamatsu Photonics, Hamamatsu, Japan)) and Adaptive optics unit with MicAO module from Imagine optic for 3D imaging. SR Apochromat TIRF 100x NA = 1.49 oil immersion was used to provide the highest quality point spread function. Resolution limits of the instrument were define as: lateral (x,y) = 20 to 30 nm and axial (z) = 50 to 60 nm. Excitation was performed using 647 nm (125 mW at the fiber tip) and 561 nm (70 mW at the fiber tip) lasers acquiring 20,000 frames per image with an acquisition time of 10 ms per frame. Imaging was performed in dSTORM super-resolution buffer (Abbelight, Cachan, France) and TetraSpeck Microspheres, and 0.1 μm beads (Thermo Fisher Scientific, T7279) were used for calibration.

## STORM analysis

Analysis and measurement steps were performed using ThunderSTORM [81] and ZOLA [82] plugins of ImageJ. Custom python scripts were developed for specific tasks. The pipeline and the parameters used in the different steps are specified in S11 Fig.

Colocalization for STORM experiments was performed adapting the Coordinate-based colocalization analysis (CBC) [83]. Briefly, in CBC colocalization, between each point from channel A is compared to points in channel B in ray a (100 nm) versus ray b where b is larger than a (300 nm). The resulting measurement is normalized for volume difference. Each localization of the observed protein population is attributed to an individual colocalization value ($C_A$) calculated as Spearman correlation coefficient between the linearized distribution functions of both protein populations in the neighborhood of the very localization. $C_A$ range from −1 to 1. We selected as colocalization events points with a $C_A > 0.8$ [83].

## Quantification and statistical analysis

Statistical analysis was performed using Python and GraphPad Prism. Data are presented as mean centered and the standard deviation. All experiments were repeated with at least 3

independent biological replicates, unless stated otherwise. The statistical test used to evaluate significance is the Welch $t$ test. Statistical significance in the figures is denoted as follows: ns: $p > 0.05$, $^*$: $p < 0.05$, $^{**}$: $p < 0.01$, $^{***}$: $p < 0.001$.

## Code availability

All custom scripts used in this study are available from the corresponding author upon reasonable request.

## Supporting information

**S1 Table. Selected candidate gene list.** Top 10 candidates identified in PMP screening and sorted according to HaSAPPy score. Number of I.I. and D.I. detected in control and selected samples are provided for each gene. Subcellular localization of corresponding proteins is indicated (Cell. comp.). D.I., disrupting insertion; I.I., independent insertion; PMP, purmorphamine.
(PDF)

**S2 Table. OLIG2 and NKX6.1 positive cells in neural tube sections of *Tmed2*$^{-/-}$ and WT embryos.** Six neural tube sections of E9.0 WT and E10.5 *Tmed2*$^{-/-}$ embryos were analyzed and cells positive for the OLIG2 and NKX6.1 were counted. Percentage of OLIG2 relative to NKX6.1 positive cells is provided for each section. WT, wild-type.
(PDF)

**S1 Text. Supporting information references.**
(PDF)

**S1 Fig. SMO supports survival of ESCs after dissociation. (A)** Generation of *Smo*$^{-/-}$ ESCs by CRISPR/Cas-9 gene editing. Scheme showing *Smo* gene structure and transcripts variants. gRNAs were designed against the first exon of both transcripts. Sequence of the locus is shown for WT cells and for 2 clones where Cas-9–dependent NHEJ repair introduced frameshift mutations in the *Smo* coding region. Both mutations lead to a frameshift in the *Smo* coding sequence compromising protein expression. **(B)** Proliferation of WT and *Smo*$^{-/-}$ ESCs in ES and Ndiff+2i media. **(C)** GLI transcription is not affected by *Smo* deletion in ESCs. *Ptch1* mRNA levels were measured by qPCR ($n = 3$, biological replicates). **(D)** GLI transcription remains unchanged in ESCs treated with HH pathway targeting compounds. ESCs were treated for 48 hours with SHH (100 ng/ml), SAG (0.5 μM), PMP (1 μM), Cyclopamine-KAAD (1 μM), and SANT-1 (10 μM) as indicated, and *Gli1* mRNA levels were measured by qPCR. Samples do not show statistically significant differences ($n = 3$, biological replicates). **(E, F)** Chemical inhibition of GLI activity does not affect ESCs survival. (E) Survival of ESCs after 48 hours with or without addition of the GLI inhibitor GANT61 (1μM). (F) Effect of GANT61 treatment on *Gli1* expression. Expression of *Gli* mRNA with or without GANT61 (1μM) treatment in ESCs ($n = 3$, biological replicates) (left). Induction of *Gli* mRNA by SHH with or without GANT61 (1 μM) treatment in NIH-3T3 ($n = 3$, biological replicates) (right). **(G, H)** Frequency of ciliated ESCs. (G) Immunofluorescence showing ciliated cells in an ESC colony. ARL13B marker was used to stain cilia (indicated with an arrow). Scale bar = 10 μm. (H) Plot of the percentage of ciliated cells over the total number of cells. Each dot represents the percentage of cells with cilia detected in an ESC colony ($n = 25$). The data underlying all the graphs shown in the figure are included in the S1 Data file. ESC, embryonic stem cell; gRNA, guide RNA; HH, hedgehog; NHEJ, nonhomologous end joining; PMP, purmorphamine; qPCR, quantitative PCR; SAG, smoothened agonist; SHH, Sonic hedgehog; SMO,

Smoothened; WT, wild-type.
(PDF)

**S2 Fig. High concentrations of SAG and PMP induce cell death by sequestering SMO protein. (A)** *Gli* transcriptional activity in NIH-3T3 cells treated with compounds shown in Fig 2B. *Gli1* mRNA was measured by RT-qPCR. **(B)** Effects of compounds indicated on survival of NIH-3T3 cells. NIH-3T3 cells were treated for 48 hours with SHH (50–500 ng/ml), SAG (1–5 μM), PMP (2.5–10 μM), Cyclopamine-KAAD (1–5 μM) or SANT-1 (10–50 μM). Survival rate is normalized to DMSO treated sample. **(C)** Dissociation-induced apoptosis is prompted by high PMP concentrations. Survival of ESCs treated for 48 hours with different PMP concentrations (2 nM—20 μM). Red dotted line highlights IC50 at 2.5 μM. **(D, E)** High concentrations of PMP repress GLI activity and HH signaling. (D) RT-qPCR of *Gli1* mRNA in NIH-3T3 cells treated with increasing concentrations of PMP (2 nM to 40 μM) and normalized to untreated samples. Dotted lines mark PMP EC50 (green, at 30 nM) and IC50 (red, at 4 μM). (E) RT-qPCR of *Gli1* mRNA in *Ptch1*$^{-/-}$ MEFs treated with increasing concentrations of PMP (2 nM to 40 μM) and normalized to untreated samples. Red dotted line highlight IC50. **(F)** SAG and PMP cytotoxic effects are reduced in *Smo*$^{-/-}$ ESCs. Survival of *Smo*$^{-/-}$ and control ESCs treated for 48 hours with SAG (2.5 to 5 μM) and PMP (5 to 10 μM). The effect on 2 independent clones is shown. Asterisks denote statistical significance for difference from the DMSO treated samples. **(G)** PMP induces blebbing and death after dissociation of ESCs. On the left, ESC morphology 24 hours after dissociation and plating on Matrigel (upper panels, pretreated with PMP for 24 hours; lower panels, pretreated with PMP and with ROCKi). Cells showing membrane blebbing (arrow), and apoptotic bodies (asterisk) are indicated. Images at 1, 3, and 6 hours after plating are shown on the right. **(H)** Survival of ESCs treated for 48 hours with PMP with or without the addition of ROCKi. **(I)** Phosphorylated MYL2 (MYL2-P) distribution in WT and *Tmed2* mutant ESCs treated with or without PMP for 24 hours. Bars represent 5 μm. **(J)** Constitutive active *Rac1*(CA) confers PMP resistance. Two independent ESCs lines expressing the Rac1(CA)-IRES-GFP cassette were treated with PMP for 48 hours. Cell survival was normalized to DMSO treated WT ESCs. **(K)** Immunoblot analysis showing expression of Rac1(CA)-IRES-GFP in ESCs used in S2J Fig. **(L)** Western analysis showing SMO–HA expression levels in the ESCs clones used in 2E Fig. Actin is blotted as loading control. The data underlying all the graphs shown in the figure are included in the S1 Data file. ESC, embryonic stem cell; HA, hemagglutinin; MEF, mouse embryonic fibroblast; PMP, purmorphamine; RT-qPCR, quantitative reverse transcription PCR; SAG, smoothened agonist; SMO, Smoothened; WT, wild-type.
(PDF)

**S3 Fig. Genetic screen in haploid ESCs for PMP resistance identifies Anoikis modulators. (A)** Selection of ESCs resistant to PMP. Infected (3 independent experiments) and control cells were treated for 12 days with PMP. ESCs were passed and counted every 2 days. The graph shows the percentage of the counted cells relative to the number of cells plated on day 0. The dashed line indicates a baseline of irradiated MEFs used for ESC culture. **(B)** Depiction of insertion numbers genome wide (gray) and in gene transcription units (black), for control and PMP selected samples. **(C)** Chromosomal distribution of insertions in control (blue, above) and selected (red, below) samples are shown. **(D, E)** Two-dimensional plots of fold enrichment of I.I. and D.I. (left panel), I.I. and Bias (central panel), and D.I. and Bias (right panel) of genes during selection. Top hits genes involved in anchorage independent growth (D) or related to the ER–Golgi compartments (E) are marked in red and annotated. **(F)** Summary model showing the role of selected candidates in the Anoikis cascade: Myh9 [41,84], Phactr4 [85], Tpm3 [86], Rock2 [87], Mypt1 [88,89], and Dapk3 [89]. The data underlying all the graphs shown in

the figure are included in the S1 Data file. ER, endoplasmic reticulum; ESC, embryonic stem cell; I.I., independent insertion; MEF, mouse embryonic fibroblast; PMP, purmorphamine.
(PDF)

**S4 Fig. Identification of *Tmed2* as a new modulator of HH signaling. (A)** Number of insertions in genes of the p24 family that were detected in selected and control samples. **(B)** Generation of *Tmed2*$^{-/-}$ and *Tmed10*$^{-/-}$ ESCs lines using CRISPR/Cas-9 technology. On the top schematic representation of the *Tmed2* (above) and *Tmed10* (below) gene locus with exons (numbers within boxes), gRNA position (red), and MscI (*Tmed2*) and FspI (*Tmed10*) restriction sites that were used to identify the gene edited clones are indicated. Sequence of the locus is shown for WT cells and for 2 clones where Cas-9 dependent NHEJ repair introduced frameshift mutations in the *Tmed2* and *Tmed10* coding region. Blue and black colors mark nucleotides in exonic and intronic regions, respectively. **(C, D)** *Tmed2* and *Tmed10* mutations are compatible with ESC self-renewal. Plots show relative growth rates of 2 independent *Tmed2*$^{-/-}$ and *Tmed10*$^{-/-}$ ESC clones compared to parental ESCs. Dots show individual measurements, error bars represent standard deviation (WT; $n$ = 5). The data underlying all the graphs shown in the figure are included in the S1 Data file. ESC, embryonic stem cell; gRNA, guide RNA; HH, hedgehog; NHEJ, nonhomologous end joining; WT, wild-type.
(PDF)

**S5 Fig. *Tmed2* is a negative regulator of HH signaling in neuronal differentiation. (A–C)** *Tmed2* depletion increases GLI transcriptional activity in NIH-3T3 cells. NIH-3T3 cells were transfected with either a siRNA targeting *Tmed2* (si*Tmed2*) or with a negative control siRNA (siC-) and treated with SHH (100ng/ml) for 6 or 24 hours. (A) RT-qPCR analysis of *Ptch1* transcription; data are expressed as fold increase relative to siC-transfected cells. (B) Plot of the percentage of ciliated cells over the total number of cells. By immunofluorescence imaging, ARL13B marker was used to stain cilia in cells that were depleted for *Tmed2* (black bars) or not (gray bars) as indicated. (C) Quantification of SMO recruitment to the primary cilia after 6 hours of SHH treatment in cells that were depleted for *Tmed2* or not as in panel D. Number of cells positive for the SMO staining in primary cilia was normalized to the total number of counted cells. **(D)** TMED2 expression in the neural tube. Representative image of immunostainings of TMED2 and RCAS1 used to visualize the Golgi apparatus in neural tube sections of E9.5 embryos. Square indicates area magnified on the right. Scale bar = 10 μm. **(E)** Immunostaining of TMED2 (red) and the Golgi marker RCAS1 (green) in neural tube sections of *Tmed2*$^{-/-}$ E10.5 mouse embryos. Scale bar = 20 μm. The data underlying all the graphs shown in the figure are included in the S1 Data file. HH, hedgehog; RT-qPCR, quantitative reverse transcription PCR; SHH, Sonic hedgehog; siRNA, small interfering RNA; SMO, Smoothened.
(PDF)

**S6 Fig. Mutation of *Tmed2* affects neural tube pattering. (A)** Neural tube sections of E10.5 *Tmed2*$^{-/-}$, E9.0, E9.5, and E10.5 control embryos stained for the ventral markers OLIG2 and NKX2.2. Scale bar = 20 μm. **(B, C)** Neural tube sections of E10.5 (B) and E9.0 control and E10.5 *Tmed2*$^{-/-}$ (C) embryos were stained for the ventral markers OLIG2 and NKX6.1. Scale bar = 20 μm. **(D)** Neural tube sections of E9.0 control and E10.5 *Tmed2*$^{-/-}$ embryos were stained for the markers DBX1 and NKX6.1. Scale bar = 20 μm. **(E)** Percentage of DBX1 (upper) and NKX6.1 (lower) expressing NPCs relative to total NPCs in neural tube sections of E9.0 control and E10.5 *Tmed2*$^{-/-}$ embryos. Asterisks denote statistical significance for difference between indicated samples. **(F)** PAX7 expression in a neural tube section of an E10.5 control embryo. Scale bar = 20 μm. The data underlying all the graphs shown in the figure are

included in the S1 Data file. NPC, neural progenitor cell.
(PDF)

**S7 Fig. SMO interacts with TMED2 in the Golgi compartment. (A)** Model showing the cellular compartments and their respective markers for analyzing SMO distribution. **(B)** Immunofluorescence showing costaining of C-term SMO–HA (red) with ERP72, RCAS1, SYNTAXIN-6, RAB7, and EEA1 (green). Scale bar = 10 μm. **(C, D)** N-term HA–SMO WT (upper panels) and A1 mutant (lower panels) costaining with ERP72 (C) and RCAS1 (D). Scale bar = 10 μm. **(E)** Western analysis showing HA–SMO WT and A1 expression levels in the ESCs clones. Actin is blotted as loading control. **(F–H)** SR microscopy distribution of SMO–HA (red) and ERP72 (F), RCAS1 (G) and RAB7 (H) (green) in the SR experiments shown in Fig 6A–6C. Scale bar = 5 μm. **(I)** Cellular localization of SMO–HA (red) and EEA1 (green) in SR experiments. Colocalizing events are shown in independent plots and labeled in gray; scale bar = 5 μm. ESC, embryonic stem cell; HA, hemagglutinin; SMO, Smoothened; SR, super-resolution; WT, wild-type.
(PDF)

**S8 Fig. Generation of SMO endogenously tagged ESCs by CRISPR/Cas-9. (A)** Scheme showing the CRISPR/Cas-9 strategy used to introduce a 3xHA epitope tag into the endogenous *Smo* gene locus in ESCs. Location of primers and restriction sites used for screening positive clones are marked. **(B)** PCR analysis of WT and edited clones using primers flanking the edited region (top left) and specific for the edited region (bottom left). PCR products obtained using the external primers were digested with XhoI (present in WT cells, top right) and with EcoRI (introduced with the 3xHA epitope). All analyzed clones are characterized by the integration of the 3xHA sequence. The B3 clone shows a heterozygous genotype with just a single allele edited. **(C)** Sequence of the targeted locus is shown for WT cells and for A10 and B11 clones. **(D)** Immunofluorescence showing HA staining in WT and 3xHA–SMO ESCs. Scale bar = 10 μm. **(E)** Immunofluorescence showing colocalization of HA staining (red) with TMED2 (green) in WT and 3xHA–SMO ESCs. Cells were subjected after fixation to an antigen retrieval protocol specific for the detection of ER–Golgi resident proteins (see Materials and methods for details). Scale bar = 10 μm. ER, endoplasmic reticulum; ESC, embryonic stem cell; HA, hemagglutinin; SMO, Smoothened; WT, wild-type.
(PDF)

**S9 Fig. Effects of the *Tmed2* mutation on cellular organelle distribution.** Immunofluorescence showing ERP72 **(A)**, RCAS1 **(B)**, RAB7 **(C)** and SYNTAXIN-7 **(D)** distribution in control (left panels) and *Tmed2*$^{-/-}$ (right panels) cells. Scale bar = 5 μm in (A), (C), and (D); scale bar = 4 μm in (B).
(PDF)

**S10 Fig. SR distribution of SMO–HA in cells treated with SHH or mutated for *Tmed2*. (A–C)** SR SMO colocalization with (A) ERP72, (C) RCAS1 and (D) RAB7 in control (untreated and SHH treated) and *Tmed2*$^{-/-}$ NPCs. SMO–HA cellular distribution (red) was compared to organelle marker staining (green) in dual-color 3D-STORM experiments. Colocalizing events are shown in independent plots and labeled in gray; scale bar = 5 μm. Magnified area are shown on the left; scale bar = 2 μm. **(B)** Plot showing the percentage of SMO–HA colocalizing events relative to ERP72 localized in perinuclear or peripheral area upon SHH treatment and *Tmed2* mutation. The data underlying all the graphs shown in the figure are included in the S1 Data file. NPC, neural progenitor cell; SHH, Sonic hedgehog; SMO, Smoothened; SR, super-resolution.
(PDF)

**S11 Fig. TMED2 regulates SMO abundance at the PM in a SHH-dependent manner. (A)**
SR SMO colocalization with EEA1 in control (untreated and SHH treated) and $Tmed2^{-/-}$
NPCs. SMO–HA cellular distribution (red) was compared to organelle marker staining
(green) in dual-color 3D-STORM experiments. Colocalizing events are shown in independent
plots and labeled in gray; scale bar = 5 μm. **(B, C)** $Tmed2$ regulates SMO localization at the
PM. Cells were treated with NHS-SS-Biotin (Biotin) to label PM proteins, and with SHH as
indicated. Western analysis of proteins present at the PM and input before purification (1/50
of pull-down) is shown. **(D)** The GPCR FZD2 binds to TMED2 in 293T cells. Western analysis
of co-immunopurification of endogenous TMED2 with SMO–HA and FZD2-HA transfected
in 293T cells (top) and input (1/25 of IP, bottom). **(E)** FZD2 secretion is not increased by the
$Tmed2$ mutation. WT and $Tmed2^{-/-}$ ESCs overexpressing FZD2-HA were treated with
NHS-SHHS-Biotin (Biotin) to label PM proteins. Western analysis of proteins present at the
PM and input before purification (1/50 of pull-down) is shown. **(F)** Western analysis of co-
immunopurification of endogenous TMED2 with SMO–HA (top) and input (1/25 of IP, bot-
tom) in NSCs overexpressing C-term HA tagged SMO. Cells were treated with recombinant
SHH for the indicated amount of time. Actin is shown as a loading control. HA, hemaggluti-
nin; NPC, neural progenitor cell; NSC, neural stem cell; PM, plasma membrane; SHH, Sonic
hedgehog; SMO, Smoothened; SR, super-resolution; WT, wild-type.
(PDF)

**S12 Fig. Pipeline of the dual color 3D-STORM analysis.**
(PDF)

**S1 Raw Images. Original scan images.**
(PDF)

**S1 Data. Numerical values used for plots and statistical analysis in Figs 1D, 1E, 1G, 1H, 1I,
2B, 2E, 2I, 3F, 4A, 4F, 5B, 5C, 6E, 7F and 7H and S1B, S1C, S1D, S1E, S1F, S1H, S2A, S2B,
S2C, S2D, S2E, S2F, S2H, S2J, S3A, S3B, S3C, S3D, S3E Fig, S4A, S4C, S4D, S5A, S5B, S5C,
S6E, and S10B Figs.**
(XLSX)

# Acknowledgments

We thank M. de Palma for providing the lentiviral gene trap vector; G. Thinakaran for sharing
the TMED10 antibody; M. Kruse and M. Torres for helping generating cellular models and
reagents; R. Freimann for assistance with cell sorting; D. Pintosi and the ScopeM facility for
assistance with STORM imaging and analysis; and M. Ghidinelli, U. Suter, and V. Taylor for
help with reagents and discussions.

# Author Contributions

**Conceptualization:** Giulio Di Minin, Anton Wutz.

**Data curation:** Giulio Di Minin, Markus Holzner, Alice Grison, Charles E. Dumeau.

**Formal analysis:** Giulio Di Minin, Markus Holzner, Alice Grison, Charles E. Dumeau, Loydie
A. Jerome-Majewska.

**Funding acquisition:** Giulio Di Minin, Anton Wutz.

**Investigation:** Giulio Di Minin, Markus Holzner, Alice Grison, Charles E. Dumeau, Wesley
Chan, Asun Monfort, Anton Wutz.

**Methodology:** Giulio Di Minin, Markus Holzner, Alice Grison, Charles E. Dumeau, Wesley Chan, Asun Monfort.

**Project administration:** Giulio Di Minin, Anton Wutz.

**Resources:** Giulio Di Minin, Wesley Chan, Asun Monfort, Loydie A. Jerome-Majewska, Henk Roelink, Anton Wutz.

**Software:** Giulio Di Minin.

**Supervision:** Giulio Di Minin, Anton Wutz.

**Validation:** Giulio Di Minin.

**Visualization:** Giulio Di Minin.

**Writing – original draft:** Giulio Di Minin, Loydie A. Jerome-Majewska, Henk Roelink, Anton Wutz.

**Writing – review & editing:** Giulio Di Minin, Anton Wutz.

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
