## [Editor Report · Decision Letter 0]

20 Jul 2021

Dear Dr Wutz, 

Thank you for submitting your manuscript entitled "TMED2 acts upstream of SMO to restrict it to the ER-Golgi" for consideration as a Research Article by PLOS Biology. Thank you also for your patience as we completed our editorial process, and please accept my apologies for the delay in providing you with our initial decision.

Your manuscript has now been evaluated by the PLOS Biology editorial staff as well as by an academic editor with relevant expertise and I am writing to let you know that we would like to send your submission out for external peer review.

Please re-submit your manuscript within two working days, i.e. by Jul 22 2021 11:59PM.

Kind regards,

Ines

--

Ines Alvarez-Garcia, PhD

Senior Editor

PLOS Biology

---

## [Decision Letter · Decision Letter 1]

13 Sep 2021

Dear Dr Wutz,

Thank you for submitting your manuscript entitled "TMED2 acts upstream of SMO to restrict it to the ER-Golgi" for consideration as a Research Article at PLOS Biology. Thank you also for your patience as we completed our editorial process, and please accept my apologies for the delay in providing you with our decision. Your manuscript has been evaluated by the PLOS Biology editors, an Academic Editor with relevant expertise, and by three independent reviewers.

As you will see, the reviewers find the conclusions interesting and novel, but they also raise several concerns that need to be addressed with additional experiments to confirm the results. Both Reviewers 1 and 3 raise the fact that the Tmed2/Smo interaction might not be specific and that there might be some off-target effects in the Tmed mutant, thus you will need to perform experiments to demonstrate specificity. In addition, Reviewer 1 will need to be convinced that release of SMO from the Golgi is a key event that is stimulated by HH pathway induction. Reviewer 2 also asks for several experiments to bolster the main conclusions, along with quantification of some experiments and clarifications.

In light of the reviews (attached below), we will not be able to accept the current version of the manuscript, but we would welcome re-submission of a much revised version that takes into account all the reviewers' comments. We cannot make any decision about publication until we have seen the revised manuscript and your response to the reviewers' comments. Your revised manuscript is also likely to be sent for further evaluation by the reviewers.

We expect to receive your revised manuscript within 3 months. 

**IMPORTANT - SUBMITTING YOUR REVISION**

*Re-submission Checklist*

*Published Peer Review*

*PLOS Data Policy*

*Blot and Gel Data Policy*

Sincerely,

Ines

--

Ines Alvarez-Garcia, PhD

Senior Editor

PLOS Biology

Reviewers’ comments

Rev. 1:

This manuscript 1) outlines a screen performed to identify new modulators of Sonic Hedgehog (SHH) signaling, and 2) examines functionality of a hit from the screen (TMED2) in regulation of SHH signal output in vertebrates. The authors provide cell biological and genetic data suggesting that TMED2 plays a negative regulatory role for control of SHH by preventing exit of the signal transducing component Smoothened (SMO) from the Golgi. Identification of TMED2 and a related protein TMED10 as modulators of SHH signaling will be of interest to some in the field. However, concerns about experimental design and data interpretation limit enthusiasm for the work and make it unsuitable for publication in its current form. Several experiments lack appropriate controls, validation of the screen is insufficient, and conclusions are over-stated. Primary concerns are that the screen is not well validated, several experiments lack controls, and results are over-interpreted. It is also a confusion for this reviewer that the authors screen for 'noncanonical' SHH signaling activities, then investigate a hit from this screen by examining canonical signaling.

Specific points to be addressed are listed below.

The authors observe that embryonic stem cells (ESCs) that lack SMO have a slower growth rate and undergo apoptosis at a higher rate than control cells. They hypothesize that SMO plays a role in preventing death of pluripotent cells. The authors investigate whether this is a canonical (GLI dependent) or noncanonical (GLI independent) signal by looking at qPCR of SHH gene targets Gli1 and Ptch (Figure 1). Given that Gli1 and Ptch are unaffected by Smo loss or small molecular modulators in ESCs, the authors conclude that defective canonical signaling is not leading to cell death. However, Gli1 and Ptch may not be reliable indicators of all the genes induced by SHH signaling. Phenotypes could be result from altered regulation of other target genes. What happens if you do similar treatments in the presence of a Gli antagonist (GANT)?

Figure 1E-F, S1C - how many replicates are represented by the data? Do the dots equal three biological replicates or technical replicates? Please provide this information in the figure legends.

Figure 1G - The authors should indicate whether cyclopamine treatment decreases survival. This experiment is needed to support that PTX effects are due to compromised G alpha i (Gi) signaling downstream of SMO. What happens if you pre-treat with SHH? Does PTX have an effect in Smo -/- cells?

Figure 2C - The authors are using SMO staining in the 'periphery' to estimate membrane localization. Performing extracellular staining on non-permeabilized cells would be a clearer way to visualize plasma membrane SMO.

Figure 2D -The authors state that they see less SMO on the membrane after treatment with high SAG or PMP. They make the assumption that cell surface SMO is reduced because they do not see as much higher-molecular weight glycosylated SMO following drug treatment. Glycosylation is not a clear indicator of plasma membrane localization. The results provided do not properly demonstrate this. Cell surface biotinylation would be more informative biochemical technique to use.

The authors use super-physiological concentrations of SAG and PMP in the screen and validations, so there is concern they are validating off-target effects.

Figure 3E - differences in SMO localization minus and plus Tmed2 are unconvincing.

Figure 4A - TMED2 depletion is only having an effect in the presence of SHH? What happens following Tmed2 knockdown in the absence of SHH? Does basal activity come up? If not, is it a true negative modulator?

Page 12, line 11 - the statement that TMED2 is regulating canonical signaling based upon the results that SMO movement into primary cilia is unaffected by Tmed2 knockdown doesn't make sense. SMO entry into the primary cilium controls canonical signaling. Perhaps signaling to GLI is impacted, but the authors should not use the phrase 'canonical signaling' based on this result. Also, why is something that you pulled out as being part of the noncanonical response (based upon your screening strategy in ES cells) affecting GLI?

Figure 4C - this panel is not very informative because there is no control with TMED2 mutant neural tubes to support staining is specific.

Figure 4E - need to show the TMED2 staining to confirm the EBs are really knockout tissue. Also, please show TMED2 knockouts without SHH stimulation. Again - it's unclear if changes are only occurring in the presence of SHH or if basal pathway signaling is enhanced in the absence of TMED2. From panel G, it looks like signaling is not increased in the absence of SHH. What might this mean?

The labeling on panel 4E is unclear. What is the genotype of the EB that is one up from the bottom?

Figure 5 - Embryos of different developmental stages were compared between Tmed-/- animals and WT embryos to account for developmental delays/size differences. It is not appropriate to make this comparison because lots of patterning changes occur between E10.5 (mutant) and E9.0-9.5 (WT examples). This leads to concern that conclusions based on differences between WT and mutant neural tubes are not valid.

Page 14, lines 11-13: The authors state that they 'note that the increase in HH signaling in the absence of TMED2 is different from ectopic activation caused by absence of negative modulators'…but don't comment on why this might be the case.

Figure 6: For colocalization experiments with ER, Golgi and RAB markers, statistical analysis should be performed. Can the coordinate based colocalization (CBC) test results that are mentioned in line 24, page 15 be subjected to statistical analysis? Quantification provided in 5G is not sufficient - also, need clarification in the text as to what this panel shows. "Quantification showed that 40% of TMED2 is tightly associated with SMO" is not an informative statement. Can Pearson's colocalization coefficients be determined and statistically analyzed?

Figure 7G: Cell surface biotinylation of SMO is shown in Tmed2 -/- cells compared to WT cells following SHH treatment. Functional experiments (Figure 4A, for example) revealed that Tmed2-/- cells only showed a difference from wild type following SHH treatment. Why are the experimental conditions/conclusions inconsistent?

For changes in SMO molecular weights, the authors speculate that SMO is differentially modified in control vs. Tmed2 null cells, but there is no discussion of why this might be. Have the authors considered that SMO is skipping the Golgi in Tmed2 null cells? This has previously been reported for SMO-M2 (PMC3044124). Or, as the authors propose, it may be due to SMO spending less time in the Golgi in the absence of Tmed2.

The authors looked at two other cell surface proteins (N-cad and EGFR) and saw that their cell surface localization was unaffected by TMED2 loss. From this result, they conclude that TMED2 effects on cell surface localization are 'specific' to SMO. I don't think the conclusion of specificity can be made from looking at only two additional cell surface localized proteins. SMO is probably one of several cell surface-directed proteins that is influenced by TMED2. So, I recommend toning down the specificity argument.

The authors speculate that release of SMO from the Golgi is a key event that is stimulated by HH pathway induction. This hypothesis stems from their observation that the majority of their SMO signal is evident in the ER and Golgi compartments. However, the localization studies are conducted using over-expressed SMO, and it's unclear whether these conclusions would apply to protein expressed at endogenous levels. Over-expressed cell surface proteins typically pile up along the secretory pathway, and may not represent the trafficking behavior of a protein expressed at physiological concentration. Thus, I am not persuaded by the argument.

Minor point: Page 4 lines 11-13 of introduction: The authors state the nature of the SHH response (canonical or noncanonical) is determined prior to SMO cycling through the primary cilium. However, there may be biased agonists that control the SMO effector route selection, which would represent a trafficking independent route selection process. The authors should also consider that SMO glycosylation may impact the GLI vs. non-GLI response (PMC4546403).

Rev. 2:

Minin and colleagues present some novel data on a potential role of TMED2 in the regulation of Hedgehog signaling through limiting SMO trafficking to plasma membrane. The authors firstly demonstrated that SMO is critical for ESCs survival and this function is independent of transcription factor GLIs. Based on the finding that SMO GPCR function is important for ESC survival, they then identified TMED2 mutant enhances SMO GPCR function and elucidated its mechanism. The concept for this paper is novel and the conclusion is supported by their data. There are still some concerns regarding the manuscript.

1) Fig.2 and Fig. S2 provide much information in the effect of PMP/SAG on GLI activation and SMO-mediated ESC survival. Several confusing points in Fig. 2 and Fig. S2 need to be addressed. In Fig. S2F, two independent SMO-/- clones showed different results upon high dose PMP / SMO treatment. Statistical analysis is needed here, as this result is critical for understanding whether PMP works at the upstream of SMO or not. In Fig S2H and Fig S2I, it is interesting that ROCK inhibition and Rac1 (CA) overexpression can only rescue half (but not all) cell death led by PMP treatment. An explanation for it may be needed. In Fig 2C and 2D, why does total amount of SMO-HA decrease dramatically by treatment of SAG or PMP, especially the lower band in PMP-treated sample in Fig 2D?

2) In page 9 line 23,"Treatment of ESCs with high PMP concentrations allows to quickly induce SMO destabilization", more data is needed to demonstrate that the decreased amount of SMO-HA by PMP treatment is due to PMP-induced SMO destabilization, but not other reason.

3) The fluorescent staining method for SMO-HA in Fig. 2C&3E, as well as ERP72, RCAS1 and RAB7 in Fig.3C need to be described.

4) In Fig. 3E, the authors showed the immunostaining result of one representative cell. However，a quantitative or semi-quantitative experiment need to be performed as the results from cell population are more credible. Western Blot is a good choice to check SMO changes in separated components of cell lysate, such as plasma membrane, cytosol and nucleus.

5) In Page 11 line 22-23, "PMP also decreased SMO levels in Tmed2 mutant ESCs but not to the extent as in control cells", the related result of Western blot should be shown.

6) In Fig. 6, why are the shape of cells and the expression pattern of SMO different with those shown in Fig.2c & 3c? Although the scale bar 5 mm has been marked, the cells in 6C and 6E look much bigger than those in Fig. 6B and 6D. Please check the original data.

Overall this is a good quality study and should be of interest to the audience.

Rev. 3:

In this manuscript Di Minin et al. investigate a novel interaction between Tmed2/10 and the essential Hh-pathway component Smoothened. They show that non-canonical Smo function is required for the survival of ES cells and that very high levels of the Hh pathway agonists SAG or Purmorphamine inhibit Smo activity and decrease ES cells survival similar to loss of Smo. Based on these observations, they perform a CRISPR-screen in haploid ES cells to identify novel genes that confer resistance to treatments with high levels of Purmorphamine, which identifies Tmed2/10 as candidate genes. The authors confirm a role of Tmed2 in the Hh pathway by characterizing gene expression patterns in neural differentiations and the in-vivo spinal cord, super-resolution imaging and extensive biochemical analysis. They propose a model in which Tmed2 retains Smo in the ER/Golgi in the absence of Hh pathway activation. This interaction is quickly released upon Hh pathway activation. As this release occurs before other well-known steps of Hh pathway activation, e.g. accumulation of Smo in the primary cilium, the authors conclude that the release of the Tmed2/Smo interaction is one of the earliest and most-upstream interactions in the Hh pathway known to date. Overall this is a strong manuscript that uses state-of-the-art techniques to address an interesting problem, the upstream regulation of the Hh pathway. The experiments are thoroughly conducted and the results are consistent. I therefore believe that this manuscript will be of interest to a wide-audience. I have couple of comments/suggestions, which I feel the authors need to address through additional experiments prior to publication, but I believe that they will be able to do so during a normal revision.

Major comments

1. Specificity of the Tmed2/Smo interaction. The authors try to show at multiple places in the manuscript that the phenotypes they observe are specific due to the interaction between Tmed2 and Smo. Yet, the phenotypes of the Tmed2 null embryos are different from those typically observed in mutants that disrupt the Hh pathway, suggesting that Tmed2 may also play a role in other cellular processes or affect other signalling pathways. This concern is further exacerbated by the fact that the actual patterning phenotype observed in the neural tube is comparably mild, e.g. the expression domain of the low-threshold target gene Nkx6.1 does not appear to be strongly expanded dorsally and the ventral target gene Olig2 is also expressed, suggesting that the Hh pathway is only mildly disrupted in these mutants. I therefore think the authors need to check if the Tmed2 mutation also affects the activity of other signalling pathways that control patterning of the spinal cord or survival of ES cells. Of specific concern here is the Wnt pathway as it relies on GPCRs closely related to Smo for its activity.

Minor comments

1. The authors claim that non-canonical Smo signalling is required for ES cell survival based on their observation that ROCK inhibitors or expression of a dominant-active Rac1 can rescue loss of Smo or Gi inhibition by Pertussis toxin. However, these treatments may just generally improve cell health or inhibit apoptosis independent of Smo activity. Do the authors observe any changes in the expression levels or activity patterns of endogenous Rock1 or Rac in Smo mutant ES cells or upon treatment with high levels of Purmorphamine?

2. The authors show that canonical signalling via Gli transcriptional activity does not work in pluripotent cells. Previous work suggested that under 2i+LIF culture conditions, similar to those used by the authors, only a small percentage of cells has primary cilia (Bangs et al. Nat. Cell Biol. 2015). I therefore wonder if there is any correlation between activation of the Gli-dependent branch of Hh signalling and the presence of primary cilia in ES cells. The authors should at least check what percentage of ES cells in their cultures exhibit primary cilia.

3. The Tmed2 rescue of high purmorphamine levels correlated with an increase of Smo at the plasma membrane. The authors argue later in the text that higher levels of Smo at the plasma membrane in Tmed2 mutants make cells hypersensitive to Shh and that treatments of NPCs with low levels of purmorphamine are sufficient for Smo relocalization from internal to external cell compartments. This leads to the questions if 1) SAG, Purmorphamine or Shh treatments cause a similar relocalization of Smo in ES cells, and 2) if the interaction between Tmed2 and Smo in ES cells is as quickly disrupted in response to Hh pathway activating compounds as the authors show later in the paper for NPCs? Addressing these questions would allow the authors to further pinpoint which aspects of the Hh pathway (upstream or downstream of Tmed2) are disrupted in ES cells.

4. Pg 13 lines 9-11: The authors say they compared the expression of Nkx2.2 and Olig2, but as far as I have seen they only show data for Olig2. They should add the data about Nkx2.2 as it might provide a more sensitive read-out due to its more ventral expression. Also, levels and duration of Shh signalling are linked in the ventral spinal cord (e.g. Dessaud et al. 2007, Balaskas et al. 2012) and the authors mention a developmental delay in the Tmed2 mutants. The authors should perform a more detailed developmental time-course of Shh target gene induction in the Tmed2 mutant mouse embryos to ensure that their e10.5 Tmed2-mutants are exposed to Shh for similar durations as their e9.0 controls.

5. Pg 14 lines 7-9: The authors only look at one marker (Pax7) here. They should add additional markers to ensure this effect is not Pax7-specific. Also, the dorsal boundary of Nkx6.1 does not seem strongly shifted in the Tmed2 mutants, suggesting that the intermediate Dbx1/2-positive domain may be strongly expanded. Is this the case?

6. Pg 18 lines 15-18: The authors look at 2 proteins (EGFR and N-Cadherin) with very different structures to Smo to show that the Tmed2 mutation has a specific effect on Smo trafficking. I would find this argument more convincing if the authors would have also looked at proteins with more similar structures, e.g. other GPCRs, to ensure that their trafficking is not perturbed.

7. Pg 19, lines 9-11: The authors state that some Tmed2 phenotypes are due to the combination with the Smo-HA transgene. For better comparison, the authors should add levels of the corresponding target genes in Smo-HA controls without Tmed2 mutation and Tmed2-mutants without Smo-HA transgene to the corresponding plots in Figure 7 and S10.

Typos and small corrections

Figure S1C: Remove "Figure 4" text box

Pg 6 line 4: The authors should clearly state if they are looking at mouse or human ES cells. This should also be mentioned in the Experimental Procedures section.

Pg 21 line 9: NCPs should be NPCs

---

## [Decision Letter · Decision Letter 2]

6 Feb 2022

Dear Dr Wutz,

Thank you for submitting your revised Research Article entitled "TMED2 acts upstream of SMO to restrict it to the ER-Golgi" for publication in PLOS Biology. I have now obtained advice from the original reviewers and have discussed their comments with the Academic Editor. 

Based on the reviews (attached below), we will probably accept this manuscript for publication, provided you satisfactorily address the remaining points raised by Reviewers 1 and 3. Please also make sure to address the following data and other policy-related requests.

In addition, we would like you to consider some suggestions to improve the title:

"TMED2 binding restricts SMO to the ER and Golgi to protect mouse embryonic stem cells from dissociation-induced cell death"

or

"TMED2 binding restricts SMO to the ER and Golgi compartments"

We expect to receive your revised manuscript within two weeks. 

*Published Peer Review History*

*Early Version*

Sincerely,

Ines

--

Ines Alvarez-Garcia, PhD,

Senior Editor,

ialvarez-garcia@plos.org,

PLOS Biology

ETHICS STATEMENT:

Thank you for providing the Ethics Statement, but please include the approval license number.

DATA POLICY:

Fig. 1D, E, G-I; Fig. 2B, E, I; Fig. 3F; Fig. 4A, F; Fig. 5B, C; Fig. 6E; Fig. 7F, H; Fig. S1B-F, H; Fig. S2A-F, H, J; Fig. S3A-E; Fig. S4A, C, D; Fig. S5A-C and Fig. S10B

Please also ensure that figure legends in your manuscript include information on WHERE THE UNDERLYING DATA CAN BE FOUND.

We require the original, uncropped and minimally adjusted images supporting all blot and gel results reported in an article's figures or Supporting Information files. We will require these files before a manuscript can be accepted so please prepare and upload them now. Please carefully read our guidelines for how to prepare and upload this data: https://journals.plos.org/plosbiology/s/figures#loc-blot-and-gel-reporting-requirements

Reviewers' comments

Rev. 1:

The revised submission is improved from the initial manuscript draft. However, some of the results remain over-interpreted, and speculation is presented as a solid conclusion on more than one occasion. The authors state that the SMO signal they investigate in ESCs is non canonical GPCR signaling. They did not conclusively show this, so the more conservative way of referring to what they are looking at is 'GLI-independent' SMO signaling since ESC viability appears to be insensitive to GANT. The PTX result +/- SMO suggests a SMO-Gi signal, but does not specifically show it is responsible for the observed effects. There is increasing evidence for a ciliary GPCR network that impacts SHH signaling, so assuming SMO is the relevant GPCR without directly showing SMO-Gi activation is risky.

The authors do show that Tmed2 is involved in SMO ER-Golgi-Plasma membrane trafficking, but I'm not convinced that the increase in plasma membrane SMO that they see in response to SHH is due to specific control of Tmed effects on SMO as is suggested in the manuscript. This is speculation.

The authors did not directly address the question from the initial review of whether cyclopamine might be nonspecifically affecting cell viability. They explained that they are using KAAD-cyclopamine, which is not what was asked.

Page 5, line 4 - reference for SMOA1 trafficking is needed.

I wanted the authors to add GANT to the panel shown in Sup Fig 1D to test for changes in Gli1 expression in the ESCs. Instead, they checked for effects on ESC viability - not the same thing as gene expression. Since they didn't show the experiment that was specifically requested, I'm left with the feeling that the result of GANT treatment on Gli1 expression was not consistent with the model the authors are trying to promote.

Figure 1H - is Ptch expressed in the ESCs? If so, SMO will not be active without pathway induction, so there is no reason to believe it would be tonically activating Gi.

Page 9, line 1 - it would be more conservative to write: "To investigate how GLI-independent SMO signaling activity is inhibited by high concentrations of PMP…" instead of specifying SMO GPCR signaling. You have not proved it is GPCR activity. The PTX effects are correlative, and do not conclusively prove that you have a Gi signal occurring downstream of SMO in the ESC system.

Figure 2E - Does over-expressed SMO accumulate on the membrane as the glycosylated form of the lower-molecular weight form?

The addition of the neutralized bodies is nice and illustrates the hyperresponsiveness to SHH of the Tmed2-/- cells. Because this result illustrates the point, I do not think it is necessary to show the neural tubes that are not age matched in the supplement. Interpretation of the data from non-age matched neural tubes is too speculative.

I'm not sure 'ciliar' is a word - 'ciliary' is clearer.

Rev. 2:

All previous concerns have been addressed in the revision.

Rev. 3:

The authors have addressed all my points raised on the previous version of the manuscript through additional experiments and by making changes to the text.

My remaining comments are minor. These are:

1.) Pg. 13, L12-14: "Tmed2-/- embryos showed a split OLIG2 domain that was shifted dorsally relative to E9.0 controls, which showed a single ventral OLIG2 domain (Supplemental Fig. S6A)." I am not entirely sure what the authors mean here with split Olig2 domain. Split by the lumen of the spinal cord into the left and right side? Comparing the first 2 embryos in Figure S6A (Tmed2- at e10,5 and WT at e9.0), these look pretty similar. Also the mentioned dorsal shift of the Olig2 domain in the mutants is not very apparent. So I feel the authors need to change the statement or need to at least better support it by a statistical quantification. Also given that the authors compare embryos from different stages of development here, it is unclear to which extent small differences in the position of DV domains are meaningful.

2.) Pg. 14, L2-3: "Consistent with this we also observed an expansion of the intermediate DBX1 expression domain in Tmed2 mutants (Fig. 5D, and Supplemental Fig. S6D).". While there seems to be a small expansion of the DBX1 domain in Fig. 5D, this is much less clear in the embryos shown in Figure S6D. Again I feel that this point should be better supported by some kind of statistical analysis.

---

## [Editor Report · Decision Letter 3]

7 Mar 2022

Dear Dr Wutz,

On behalf of my colleagues and the Academic Editor, Josh Brickman, I am pleased to say that we can in principle accept your Research Article entitled "TMED2 binding restricts SMO to the ER and Golgi compartments" for publication in PLOS Biology, provided you address any remaining formatting and reporting issues. These will be detailed in an email that will follow this letter and that you will usually receive within 2-3 business days, during which time no action is required from you. Please note that we will not be able to formally accept your manuscript and schedule it for publication until you have any requested changes.

PRESS

Sincerely, 

Ines

--

Ines Alvarez-Garcia, PhD 

Senior Editor 

PLOS Biology
